# APT: Towards Universal Scene Graph Generation via Plug-in Adaptive Prompt Tuning

**Ruikun Luo**[1234], **Changwei Gu**[1234], **Jing Yang**[5]*, **Yuan Gao**[5],
**Jieming Yang**[5], **Song Wu**[1234], **Hai Jin**[1234], **Xiaoyu Xia**[6]

[1]National Engineering Research Center for Big Data Technology and System
[2]Services Computing Technology and System Lab    [3]Cluster and Grid Computing Lab
[4]School of Computer Science and Technology, Huazhong University of Science and Technology
[5] Zhengzhou University
[6]Royal Melbourne Institute of Technology
{rkluo, gumorming, wusong, hjin}@hust.edu.cn,
{yangjinghust123, yuangaohnu, yjmlaile}@gmail.com,
xiaoyu.xia@rmit.edu.au

## Abstract

Scene Graph Generation (SGG) is pivotal for structured visual understanding, yet it remains hindered by a fundamental limitation: the reliance on fixed, frozen semantic representations from pre-trained language models. These semantic priors, while beneficial in other domains, are inherently misaligned with the dynamic, context-sensitive nature of visual relationships, leading to biased and suboptimal performance. In this paper, we transcend the traditional one-stage v.s. two-stage architectural debate and identify this representational bottleneck as the core issue. We introduce Adaptive Prompt Tuning (APT), a universal paradigm that converts frozen semantic features into dynamic, context-aware representations through lightweight, learnable prompts. APT acts as a plug-in module that can be seamlessly integrated into existing SGG frameworks. Extensive experiments demonstrate that APT achieves +2.7 improvement in mR@100 on PredCls, +3.6 gain in F@100 and up to +6.0 gain in mR@50 in open-vocabulary novel splits. Notably, it achieves this with less than 0.5M additonal parameters (<1.5% overhead) and reduced 7.8%-25% training time, establishing a new state-of-the-art while offering a unified, efficient, and scalable solution for future SGG research. The source code of APT is available at https://github.com/CGCL-codes/APT.

## 1 Introduction

Scene Graph Generation (SGG) stands as a foundational pillar in visual understanding, aiming to represent images as graphs of objects and their interrelationships in a structural manner. For years, the field has been shaped by two competing paradigms: two-stage methods, which exploit robust detector features but suffer from contextual fragmentation, and one-stage methods, which enable end-to-end learning at the expense of computational cost and relation granularity. Despite their differences, both methods share a common practice: incorporating static, fixed semantic representations—typically derived from pre-trained language models like GloVe (Pennington et al., 2014) and BERT (Devlin et al., 2019)—as semantic priors.

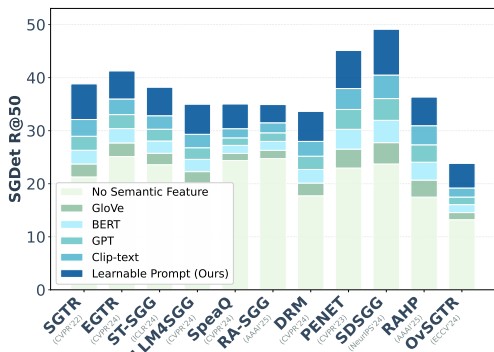

Figure 1: Performance comparison across different semantic features settings (including ours) based on various SGG methods.

While such features have proven useful in NLP Pennington et al. (2014); Devlin et al. (2019); Brown et al. (2020) and certain vision-language tasks Lu et al. (2019); Zhou et al. (2022b;a), their non-adaptive nature fundamentally limits their effectiveness in SGG, where context sensitivity, relational

---

*Corresponding author.

nuance, and role-specific semantics are paramount. As illustrated in Figure 1, we systematically compare the performance of various SGG methods under different semantic feature settings. The results reveal a consistent performance gap when models are constrained to use frozen embeddings, underscoring their suboptimal adaptability to visual relational reasoning.

The fundamental limitation lies in the rigidity of these off-the-shelf representations. Whether in a one-stage transformer or a two-stage detector, frozen word embeddings remain oblivious to visual context, incapable of distinguishing between fine-grained relations (e.g., "standing on" v.s. "walking on"), and fail to capture the semantic asymmetry between subjects and objects. For instance, the same "person" embedding remains identically frozen whether the person is riding a horse or holding a phone, which highlights a clear misalignment with the dynamic visual world. More revealingly, we visualize the feature space, as shown in Figure 2. The static semantic space collapses all 'person' instances into a single point, regardless of their diverse contexts. In contrast, the visual feature space naturally separates 'person' into clusters based on their relational context (e.g., riding, walking, etc.). This stark contrast visually demonstrates the inability of frozen representations to adapt to visual contexts and the limitation of static word embeddings in capturing relational context.

Figure 3 illustrates a two-dimensional t-SNE projection of embeddings from four mainstream pre-trained models. It also displays the distance from the central point to the farthest point within each semantic space, alongside the cumulative distribution function of pairwise distances. A progressive loosening of the feature cluster is observed when moving from GloVe to BERT and further to CLIP-text. Moreover, in Table 1, we report the silhouette score, participation ratio,

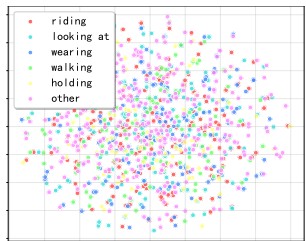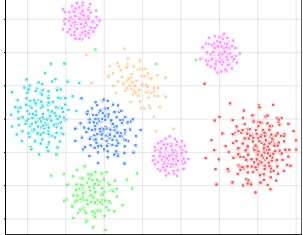

(a) Frozen GloVe Feature Space (b) Visual Context Feature Space

Figure 2: t-SNE visualization of *person* instances with visual contextual features and GloVe embeddings across different relations.

the number of principal components required to explain 90% of the variance (PCA@90), and a mutual information proxy I(embedding; predicate). This indicates that more powerful pre-trained models encode richer substructures within their representation spaces. However, this internal structure remains misaligned with the fine-grained visual-relational context required by the SGG task. This suggests that simply replacing one frozen model with another does not address the core issue of semantic rigidity. While one-stage models attempt to unify detection and relation modeling, their heavy pre-trained backbones and dense attention mechanisms incur prohibitive training costs without fundamentally solving the semantic adaptivity problem.

In light of these observations, what the community has overlooked is not the architecture, but the **representation paradigm**: the need for a lightweight, universally applicable mechanism that injects adaptive

Table 1: Diagnostics across pre-trained models.

| Embedding | Silhouette | Participation Ratio | PCA@90% | I(embedding; predicate) (bits) |
|-----------|-----------|---------------------|---------|-------------------------------|
| GloVe | 0.12 | 9.8 | 27 | 0.42 |
| BERT | 0.18 | 15.6 | 32 | 0.49 |
| GPT | 0.22 | 23.1 | 44 | 0.53 |
| CLIP-*text* | 0.29 | 48.7 | 125 | 0.57 |

semantics into any SGG framework. To this end, we transcend the one- v.s. two-stage dichotomy and introduce a new unified representation paradigm for SGG: **adaptive prompt tuning**. Rather than engineering another architecture, we propose a plug-in module that enables any existing SGG model—whether one-stage and two-stage, which are transductive, or the inductive open vocabulary setting—to dynamically modulate pre-trained semantic features in response to visual context and relational roles. Our key idea is both simple and powerful: a set of lightweight, learnable prompts that act as conditional adapters, transforming frozen language model features into context-aware representations without backpropagation through the original pre-trained backbone.

Extensive experiments validate the generality and effectiveness of our approach. When plugged into leading one- and two-stage models, our prompt module delivers consistent and significant improvements, establishing new state-of-the-art results across multiple benchmarks. Importantly, it achieves these improvements with reduced training time and minimal parameter overhead, making it highly

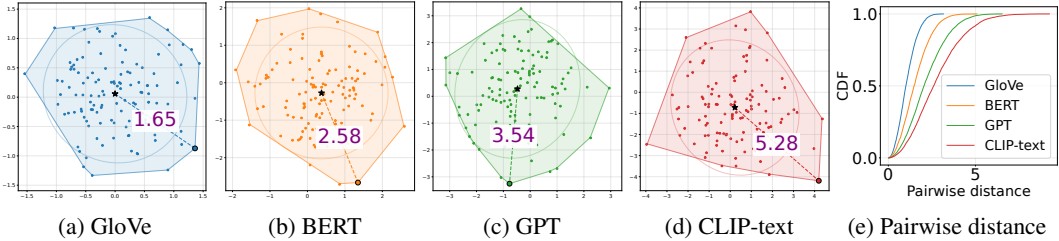

Figure 3: Two-dimensional feature distribution of four mainstream text feature models under t-SNE dimensionality reduction. The number in (a), (b), (c), and (d) represent the distance between the center point and the farthest point, while (e) shows the cumulative distribution function (CDF) of pairwise distance.

practical for compute-efficient research and applications. The main contributions of this paper are summarized as follows:

- We identify and diagnose the representational limitation of frozen, fixed semantic representations as a fundamental bottleneck in SGG, transcending architectural categories.
- We propose a lightweight, universal prompt-based representation paradigm that converts frozen semantic representations into dynamic, context-aware features, compatible with both one- and two-stage SGG frameworks.
- Extensive experiments are conducted on Visual Genome, Open Image V6, and GQA, demonstrating that APT improves mean recall (mR@100) by up to +2.7 on PredCls and boosts harmonic mean (F@100) by up to +3.6, while introducing less than 0.5M additional parameters (<1.5% overhead) and reducing training time by 7.8%–25%. In open vocabulary settings, APT achieves up to +6.0 gain on mR@50 in novel split.

## 2 RELATED WORK

Building on our diagnosis of frozen semantic representations as a fundamental bottleneck, we now review how this issue manifests across different SGG paradigms.

**Two-Stage SGG.** Two-stage methods first detect objects and then predict relations between proposed regions. Pioneering works like MOTIFS Zellers et al. (2018) built upon Faster R-CNN detectors, leveraging visual features and spatial masks to predict predicates. Subsequent efforts Tang et al. (2019) incorporated linguistic priors from pre-trained models (e.g., GloVe (Pennington et al., 2014)) to enrich object representations, while others Yang et al. (2018); Li et al. (2021) employed GNN to propagate contextual information between objects. Recent methods such as PE-Net (Zheng et al., 2023), DRM (Li et al., 2024a), and RA-SGG (Yoon et al., 2025) further explore advanced architectures to enhance relational reasoning. A key limitation is their reliance on frozen semantic representations. Whether used in label initialization or feature fusion, these fixed embeddings remain insensitive to visual context and relational nuance. Despite strong detectors, their representational capacity remains bottlenecked by non-adaptive semantic priors.

**One-Stage SGG.** In contrast, one-stage methods aim to unify detection and relation prediction within an end-to-end framework. Models such as Qpic (Tamura et al., 2021), SGTR (Li et al., 2022), EGTR (Im et al., 2024), ST-SGG (Kim et al., 2024b), LLM4SGG (Kim et al., 2024c), SpeaQ (Kim et al., 2024a) and HydraSGG Chen et al. (2025) use transformer-based architectures to directly predict relation triples from image features. These methods avoid error propagation and simplify training pipelines by jointly optimizing all components. However, these methods often inherit—and sometimes exacerbate—the problem of semantic rigidity. Many still initialize query embeddings or semantic banks using static word vectors, which cannot adapt to visual context. Moreover, their heavy reliance on self-attention over high-resolution feature maps leads to substantial computational overhead, limiting their practicality for large-scale or resource-constrained applications. The pursuit of architectural unity has thus come at the cost of both representational flexibility and efficiency.

**Open Vocabulary SGG.** Recent interest in open vocabulary SGG seeks to generalize to unseen objects and predicates. OV-SGG He et al. (2022) proposed a two-stage, prompt-based method to bridge the knowledge gap between base and novel object categories, but its prompt-template-based

fine-tuning makes it difficult to extend to other paradigms. Methods like Epic Yu et al. (2023), PGSG Li et al. (2024b), SDSGG (Chen et al., 2024a), OvSGTR (Chen et al., 2024b), SpaceSGG Xu et al. (2025), and RAHP (Liu et al., 2025) leverage large pre-trained vision-language models (e.g., CLIP (Radford et al., 2021)) for zero-shot alignment, while others employ probabilistic grounding or knowledge distillation. Yet, these methods still largely depend on frozen backbones and semantic spaces. While powerful, CLIP-based features remain generic and are not explicitly tailored to the structured and context-dependent nature of relational prediction. As a result, they often struggle with fine-grained relational reasoning and exhibit limited adaptability to downstream SGG contexts.

Our work does not propose another architecture in the one- v.s. two-stage divide, nor does it simply replace one frozen model with another. Instead, we introduce a universal plug-in module based on lightweight prompt tuning that can be seamlessly integrated into any SGG framework—whether one-stage, two-stage, or open-vocabulary.

# 3 ADAPTIVE PROMPT TUNING FRAMEWROK

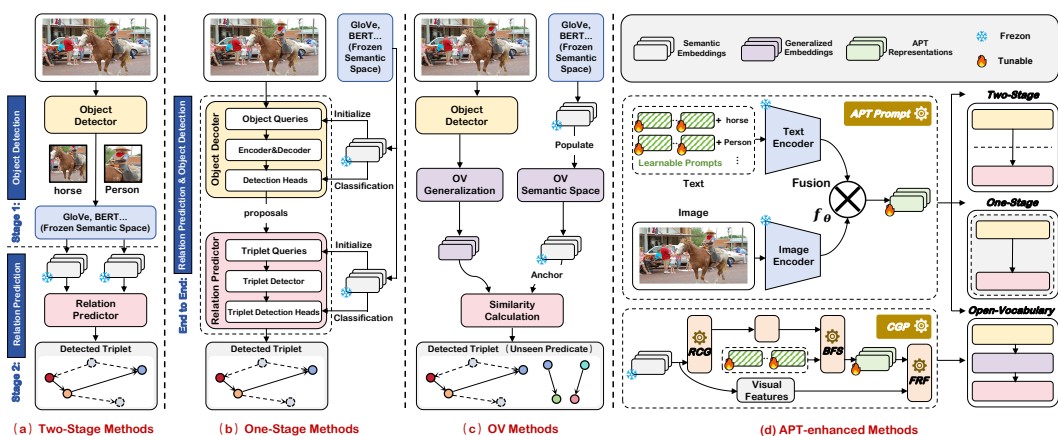

Figure 4: A comparative illustration of different SGG paradigms versus APT framework. (a) Two-stage methods suffer from fragmented context propagation and reliance on static features. (b) One-stage methods achieve end-to-end learning but at a high computational cost and with semantic rigidity. (c) Open-Vocabulary methods leverage large VL models but struggle with fine-grained relational reasoning due to generic representations. (d) Our APT framework introduces a universal plug-in module that injects adaptive semantics into any SGG backbone, enabling dynamic, context-aware feature modulation for superior performance and efficiency.

## 3.1 OVERVIEW

The **Adaptive Prompt Tuning (APT)** framework centers on lightweight, learnable prompts that adapt frozen pre-trained semantic representations into **context-aware**, **task-specific** features. As illustrated in Figure 4, APT is designed as a universal plugin that can be seamlessly integrated into both two-stage and one-stage SGG paradigms. The prompts act as conditional adapters, transforming frozen word embeddings into dynamic representations that are sensitive to visual context, fine-grained relationships, and the semantic roles of subjects and objects.

Adaptive Prompt Tuning (APT) framework is grounded in the advanced research of **continuous prompt learning** and **model adaptation**. Unlike traditional methods that discretely modify input tokens, continuous prompts introduce a set of learnable vector parameters into the model's embedding space, acting as specific instructions to guide the model's behavior Lester et al. (2021). Formally, a pre-trained model can be viewed as a function $F_\theta$ whose parameters $\theta$ are fixed after pre-training. Prompt learning aims to find an optimal prompt $P^*$ that enables the frozen model $F_\theta$ to perform best on a specific downstream task $T$:

$$P^* = \underset{P}{\arg\min} \, \mathcal{L}_T(F_\theta([P; x]))$$

(1)

APT innovatively adapts this paradigm to the multimodal context and structured prediction task of SGG. The prompt $P$ functions not as a direct prefix to a language model, but as a **feature modu-**

**lator**. It transforms generic, task-agnostic static semantic features $e_{static}$ into dynamic features $\tilde{e}$ tailored for the downstream SGG task. This process is analogous to the role of a **modem in communications**: the prompt $P$, carrying task-specific information from the visual context, "modulates" the original semantic signal, enabling it to convey information more effectively for SGG. From the viewpoint of the Information Bottleneck principle Chi et al. (2022); Yang et al. (2023); Tishby et al. (2000), APT aims to learn an optimal feature representation $\tilde{e}$ that preserves sufficient information about the object identity $c$ while maximally compressing redundant semantic information irrelevant to the current visual relation, and simultaneously incorporating relevant visual contextual information $v$. The prompt $P$ serves as the adapter achieving this "compression and injection." The objective can be formulated as:

$$\max I(\tilde{\mathbf{e}}; y) - \beta I(\tilde{\mathbf{e}}; \mathbf{e}_{static}|v, y) \tag{2}$$

In Eq. 2, $I(\cdot; \cdot)$ denotes mutual information, and $y$ is the target relation class. The first term requires $\tilde{e}$ to be informative for prediction, while the second term encourages $\tilde{e}$ to forget redundant information in $e_{static}$ given the visual context $v$ and the target $y$. The learnable prompt $P$ is optimized through training data to balance this trade-off.

## 3.2 UNIFIED PLUG-IN PROMPTS

APT operates on a simple yet powerful principle: for a given semantic concept, it employs a **lightweight, learnable prompt** $P$ to condition the frozen, static embedding $e_{static}(c)$ on the current visual context. This process can be universally described by the following equation:

$$\tilde{\mathbf{e}}(c) = f_\theta\big(\mathcal{A}(P(c), \mathbf{e}_{static}(c), \phi(\mathbf{v}))\big) \tag{3}$$

In Eq. 3, $P(c)$ is the learnable prompt for concept $c$. $\mathcal{A}(\cdot)$ is the aggregation function that reduces the prompt sequence to a single vector. $e_{static}(c)$ is the frozen pre-trained semantic embedding. $\phi(v)$ is the visual feature projector, encoding relevant visual context $v$. $f_\theta$ is a small fusion network that generates the final adaptive representation $\tilde{\mathbf{e}}(c)$

The key is that only the prompt parameters $P$, the projector $\phi$, and the fusion network $f_\theta$ are learnable. The pre-trained semantic backbone remains entirely frozen, making APT highly parameter-efficient and preventing catastrophic forgetting.

**Detection Prompt** $P_d$: This prompt is applied during the object detection phase. For each object class $c$, learnable vector $P_d(c) \in \mathbb{R}^{L_d \times D}$ is defined, where $L_d$ is the prompt length and $D$ is the feature dimension. The prompt is fused with the pre-trained semantic embedding $\mathbf{e}static(c) \in \mathbb{R}^D$ through a dedicated Multi-Layer Perceptron ($f_{\theta_{det}}$) to generate an adaptive object representation for the detection head.

**Relation Prompt** $P_r$: After objects are detected, relation prediction begins. Here, for each predicate class $r$, learnable vector $P_r(r) \in \mathbb{R}^{L_r \times D}$ is defined. This prompt is specifically designed to capture the nuances of interactions. For a subject-object pair $(s, o)$ with predicted visual features $\mathbf{v}_s$ and $\mathbf{v}_o$, their adaptive semantic features are generated and fused with visual evidence.

**Unified Relation Prompt** $P_{ur}$: Since there is no separate detection stage for one-stage paradigm, a single Relation Prompt $P_{ur}$ is sufficient and more efficient. This prompt operates on the semantic queries or label embeddings that the model uses for final predicate classification. For a potential relation with subject class $s$ and object class $o$, the model dynamically modulates their semantic embeddings. The adapted embeddings are then used by the transformer decoder for cross-attention with visual features.

In both one- and two-stage cases, the pre-trained semantic embeddings $\mathbf{e}_{static}(\cdot)$ remain **frozen**. Only the prompt parameters $P_d$, $P_r$ and the parameters of the light-weight MLPs ($f_\theta$) are learned during training. This makes our APT framework highly parameter-efficient and prevents overfitting. The same pre-trained language model can thus be shared across different SGG architectures, with the prompts specializing its knowledge for the task at hand.

## 3.3 COMPOSITIONAL GENERALIZATION PROMPTER

The Open-Vocabulary (OV) setting demands that the model generalize to unseen object and predicate compositions unseen during training. To equip APTframework with this capability, we introduce a dedicated **Compositional Generalization Prompter (CGP)**. This module is architected to dynamically synthesize context-aware semantic representations for unseen categories through

a structured, multi-stage prompting process. The CGP operates through three specialized sub-modules, which work in concert to achieve robust generalization:

**Relational Context Gating (RCG)**: This component generates role-aware prompt weights by integrating visual evidence with initial semantic cues. For a subject entity $s$ with visual feature $\mathbf{v}_s$:

$$\mathbf{w}_s = \sigma(\text{MLP}_{\text{gate}}(\text{Concat}(\mathbf{v}_s, \mathbf{e}_{\text{static}}(s)))), \tag{4}$$

The gating vector $\mathbf{w}_s$ determines the activation of prompt bases, ensuring the modulation is conditioned on the immediate visual context of each entity.

**Basis Prompt Synthesis (BPS)**: A set of learnable basis prompts $\mathbf{B} \in \mathbb{R}^{N \times L_{ov} \times D}$ serves as a repository of fundamental relational concepts. The final prompt for an entity is synthesized as a weighted combination of these bases:

$$\mathbf{P}_{\text{cgp}}(s) = \sum_{i=1}^{N} w_s^i \cdot \mathbf{B}^i, \quad \overline{\mathbf{P}_{\text{cgp}}(s)} = \text{MeanPool}(\mathbf{P}_{\text{cgp}}(s)) \tag{5}$$

To obtain a compact pooled prompt we use a normalized, token-weighted pooling with normalization ($L_b$ the basis prompt length):

$$\bar{\mathbf{p}} = \text{LayerNorm}\Big(\frac{1}{L_b} \sum_{t=1}^{L_b} \mathbf{P}_{\text{cgp}}(s)\Big) \in \mathbb{R}^D. \tag{6}$$

This allows the model to generate a virtually unlimited variety of tailored prompts from a finite set of bases, enabling compositional generalization.

**Feature Refinement & Fusion (FRF)**: This sub-module performs the final integration of the synthesized prompt, the frozen semantic embedding, and the projected visual feature:

$$\tilde{\mathbf{e}}_{ov}(s) = f_{\theta_{\text{frf}}}(\text{Concat}(\overline{\mathbf{P}_{\text{cgp}}(s)}, \mathbf{e}_{\text{static}}(s), \phi_v(\mathbf{v}_s))) \tag{7}$$

The refined feature $\tilde{\mathbf{e}}_{ov}(s)$ is context-sensitive, semantically grounded, and primed for relational reasoning with unseen concepts.

The CGP module is designed as a plug-in component that can seamlessly augment the standard Relation Prompt ($P_r$) in both two-stage and one-stage architectures. All pre-trained embeddings remain frozen. Only the basis prompts $\mathbf{B}$, the gating network, visual projectors $\phi_v$, and the fusion MLPs $f_{\theta_{\text{frf}}}$ are introduced as new learnable parameters, upholding the parameter-efficient nature of the APT framework.

By integrating **Relational Context Gating**, **Basis Prompt Synthesis**, and **Feature Refinement & Fusion**, our CGP module provides a principled and unified solution for open-vocabulary generalization, ensuring robust performance on both common and unseen compositional queries.

The overall training objective augments the loss $\mathcal{L}$ with several prompt and gating regularizers. Formally, the empirical objective—expressed as an expectation over the data distribution $\mathcal{D}$—is

$$\mathcal{L} = \mathbb{E}_{(x,y)\sim\mathcal{D}}\big[\mathcal{L}_{\text{cls}}(x,y)\big] \tag{8}$$

$$+ \lambda_p \|\mathbf{B}\|_F^2 + \lambda_{pd} \|P_{\text{det}}\|_F^2 + \lambda_{pr} \|P_{\text{rel}}\|_F^2 \tag{9}$$

$$+ \lambda_d \, \mathbb{E}_{(x,y)\sim\mathcal{D}}\big[\|\tilde{\mathbf{e}} - \mathbf{e}_{\text{static}}\|_2^2\big] \tag{10}$$

$$+ \lambda_{\text{orth}} \sum_{i<j} \big\|\mathbf{B}_i^\top \mathbf{B}_j\big\|_F^2 \tag{11}$$

$$- \beta \sum_{i=1}^{N} w_i \log w_i \tag{12}$$

$$+ \gamma \, \text{KL}\big(\mathbf{w} \,\|\, \mathbf{u}_{\text{prior}}\big) + \lambda_w \|\mathbf{W}_v\|_F^2, \tag{13}$$

where $\lambda_{\{\cdot\}}$, $\beta$, $\gamma$, and $\lambda_{\text{orth}}$ are non-negative hyperparameters; $\mathbf{u}_{\text{prior}}$ denotes an optional prior (e.g., uniform) over the gating distribution. The KL (entropy) terms serve to encourage sparsity and diversity and to penalize deviations from the prior.

# 4 EXPERIMENT

## 4.1 EXPERIMENT SETTINGS

**Datasets.** Our experiments are carried out on three publicly available benchmarks: (1) **Visual Genome (VG)** Krishna et al. (2017) comprises 150 object categories and 50 types of relations. The dataset is partitioned into 57,723 images for training, 5,000 for validation, and 26,446 for testing. (2) **Open Images V6** Kuznetsova et al. (2020) includes 288 entity classes and 30 relation categories. It provides 126,368 training images, 1,813 validation images, and 5,322 test images annotated with relational triples. (3) **GQA** Hudson & Manning (2019) contains 200 distinct entity types and 100 kinds of relations. It offers 52,623 training samples, 5,000 validation images, and 8,209 test images with scene graph annotations. We report only the results on VG due to the page limit.

**Evaluation Protocol.** The evaluation is conducted on three conventional SGG sub-tasks, including Predicate Classification (**PredCls**), Scene Graph Classification (**SGCls**), and Scene Graph Detection (**SGDet**). **PredCls** predicts the predicate classes given all ground-truth object bounding boxes and the object classes. **SGCls** aims at predicting the predicate classes given the ground-truth object bounding boxes. **SGDet** detects all entities and their pairwise predicates given an image.

**Evaluation Metrics.** We evaluate SGG models on the three metrics: (1) Recall@K (R@K) calculates the proportion of top-K predicted triplets that are in ground truth. (2) Mean Recall@K (mR@K) calculates the average recall for each predicate class, which is designed to measure the performance of SGG models under the

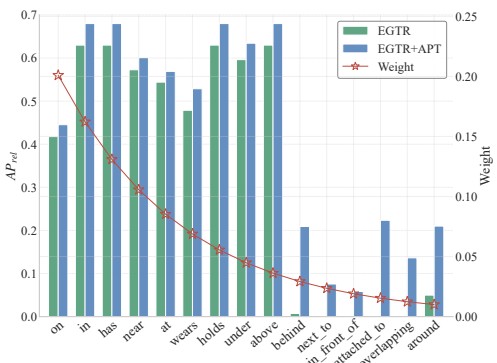

Figure 5: $AP_{rel}$ performance comparison per class. The Weight (i.e., right $y$-axis) represents the frequency ratio in the test data.

long-tailed predicate distribution Liu et al. (2019). (3) F@K calculates the harmonic average of R@K and mR@K. ST-SGG (Kim et al., 2024b) suggests that there is a trade-off between R@K and mR@K. Thus, recent works have focused on achieving greater F@K.

**Baselines.** APT is compared with methods from three categories: (1) Two-stage SGG methods, including MOTIFS (Zellers et al., 2018), PE-Net (Zheng et al., 2023), DRM (Li et al., 2024a) and RA-SGG (Yoon et al., 2025); (2) One-stage methods including SGTR (Li et al., 2022), EGTR (Im et al., 2024),ST-SGG (Kim et al., 2024b), LLM4SGG (Kim et al., 2024c), SpeaQ (Kim et al., 2024a) and HQSG (Fu et al., 2025); (3) Open Vocabulary methods including SDSGG (Chen et al., 2024a), OvSGTR (Chen et al., 2024b) and RAHP (Liu et al., 2025).

Table 2: Performance (%) of state-of-the-art SGG models with & without APT on Visual Genome Krishna et al. (2017). † denotes the results are produced using official code.

| Methods | Predicate Classification | | | Scene Graph Classification | | | Scene Graph Detection | | |
|---|---|---|---|---|---|---|---|---|---|
| | R@50/100 | mR@50/100 | F@50/100 | R@50/100 | mR@50/100 | F@50/100 | R@50/100 | mR@50/100 | F@50/100 |
| *Two-stage methods* | | | | | | | | | |
| Motif†Zellers et al. (2018)CVPR'18 | 64.6/66.0 | 15.2/16.2 | 24.6/26.0 | 38.0/38.9 | 8.7/9.3 | 14.2/15.0 | 31.0/35.1 | 6.7/7.7 | 11.0/12.6 |
| Motif+APT | 66.5/68.2 | 17.4/18.1 | 26.4/28.1 | 40.3/40.8 | 10.5/11.1 | 16.4/17.3 | 33.3/37.6 | 9.2/10.3 | 13.2/15.1 |
| PE-Net†(Zheng et al., 2023)CVPR'23 | 65.8/67.6 | 17.7/19.2 | 27.9/29.9 | 36.7/37.4 | 9.4/10.0 | 15.0/15.8 | 27.1/29.8 | 6.4/7.3 | 10.4/11.7 |
| PE-Net+APT | 67.5/69.2 | 19.3/20.5 | 29.7/31.6 | 37.2/38.4 | 10.4/10.7 | 16.2/17.1 | 28.0/31.3 | 7.1/8.6 | 11.3/12.5 |
| DRM†Li et al. (2024a)CVPR'24 | 65.8/67.6 | 17.7/19.2 | 27.9/29.9 | 36.7/37.4 | 9.4/10.0 | 15.0/15.8 | 27.1/29.8 | 6.4/7.3 | 10.4/11.7 |
| DRM+APT | 68.7/70.5 | 19.4/21.9 | 28.9/30.5 | 39.4/40.1 | 11.3/11.7 | 17.2/18.6 | 29.6/31.7 | 8.2/9.7 | 12.5/13.3 |
| RA-SGG†Yoon et al. (2025)AAAI'25 | 66.1/68.0 | 18.2/19.8 | 28.4/30.5 | 37.3/37.9 | 9.8/10.6 | 15.4/16.1 | 27.5/30.2 | 6.8/7.4 | 10.6/12.2 |
| RA-SGG+APT | 66.7/69.4 | 18.5/20.2 | 29.0/30.2 | 37.7/38.1 | 10.1/11.3 | 16.1/17.4 | 28.8/29.8 | 8.2/8.9 | 11.3/12.5 |
| *One-stage methods* | | | | | | | | | |
| SGTR†Li et al. (2022)CVPR'22 | 59.2/61.3 | 30.4/32.9 | 40.2/42.8 | 37.4/38.5 | 14.3/16.5 | 20.7/23.1 | 31.0/35.8 | 10.7/12.6 | 15.9/18.6 |
| SGTR+APT | 62.3/63.5 | 32.7/35.3 | 43.5/45.9 | 39.8/40.3 | 17.1/18.7 | 22.9/25.4 | 33.5/36.8 | 12.9/14.8 | 18.4/20.3 |
| EGTR†Im et al. (2024)CVPR'24 | 54.1/56.6 | 35.7/38.2 | 43.0/45.6 | 34.9/36.1 | 17.0/18.4 | 22.9/24.4 | 27.4/31.8 | 13.2/15.5 | 17.8/20.8 |
| EGTR+APT | 56.4/58.3 | 37.5/40.1 | 45.2/47.7 | 36.7/38.6 | 19.5/20.3 | 24.6/26.2 | 29.7/33.5 | 15.7/16.9 | 19.4/22.3 |
| LLM4SSG†Kim et al. (2024c)CVPR'24 | 62.2/64.1 | 36.2/39.1 | 45.7/48.6 | 38.2/39.1 | 20.9/22.5 | 27.0/28.6 | 26.0/30.3 | 14.4/17.1 | 18.5/21.9 |
| LLM4SSG+APT | 65.1/66.9 | 38.1/42.2 | 47.9/50.3 | 40.1/41.8 | 22.7/24.8 | 29.5/30.3 | 28.8/32.4 | 16.7/19.8 | 20.3/23.6 |
| ST-SGG†Kim et al. (2024b)ICLR'24 | 53.9/57.7 | 28.1/31.5 | 36.9/40.8 | 33.4/34.9 | 16.9/18.0 | 22.4/23.8 | 26.7/30.7 | 11.6/14.2 | 16.2/19.4 |
| ST-SGG+APT | 58.7/62.3 | 31.3/34.6 | 39.9/43.7 | 36.6/38.4 | 20.3/21.5 | 26.3/26.9 | 30.2/34.4 | 14.8/18.1 | 19.3/22.2 |
| SpeaQ†Kim et al. (2024a)CVPR'24 | 55.7/57.9 | 30.9/33.4 | 39.7/42.4 | 33.1/34.4 | 17.5/18.8 | 22.9/24.3 | 24.5/28.9 | 14.1/16.5 | 17.9/21.0 |
| SpeaQ+APT | 57.8/60.8 | 34.2/36.8 | 42.5/45.3 | 36.5/37.6 | 20.3/21.7 | 26.3/27.5 | 27.5/31.9 | 18.0/19.5 | 20.2/23.3 |
| HQSG†Fu et al. (2025)CVPR'25 | 57.6/58.9 | 32.7/34.6 | 41.5/43.2 | 35.2/36.1 | 19.2/20.4 | 25.4/27.2 | 34.1/38.3 | 16.0/20.5 | 21.8/26.7 |
| HQSG+APT | 58.7/61.2 | 35.1/37.3 | 43.3/45.4 | 37.7/38.3 | 21.9/22.7 | 26.9/28.7 | 36.5/39.9 | 18.2/21.7 | 22.6/28.2 |

## 4.2 COMPARISON WITH BASELINES ON VISUAL GENOME

As comprehensively detailed in Table 2, we evaluate the effectiveness of APT by integrating it into a diverse set of state-of-the-art SGG models, encompassing both two-stage and one-stage paradigms. The integration of APT consistently enhances the performance of all base models across the three canonical SGG tasks: PredCls, SGCls, and SGDet. This universal applicability solidifies our method's role as a powerful and general-purpose plugin for the SGG community.

The most notable improvements are observed on the mean recall (mR@K) metric, which is a more robust measure of a model's ability to predict a balanced set of predicates beyond the head classes. For instance, APT elevates the mR@100 of EGTR Im et al. (2024) by +1.9 and +1.9 on PredCls and SGCls, respectively. The striking improvement confirms that our adaptive prompts effectively mitigate the inherent bias of static features towards frequent predicates, enabling the models to perform more fairly and accurately on tail categories. This implication can be further explored by referring to Figure 5, which displays the performance $AP_{rel}$ for each class. Specifically, for the head predicates, EGTR+APT achieves competitive results. For the tail predicates, EGTR+APT significantly enhances the performance, particularly in the cases where EGTR struggles to make accurate predictions, such as **attached_to, overlapping** predicates. Furthermore, the superior F@K scores demonstrate that APT does not achieve gains in mR@K at the expense of R@K but instead fosters a more comprehensive and balanced relational understanding.

APT yields substantial gains on both two-stage and one-stage methods. This validates our core hypothesis that the limitation of frozen semantic priors is a fundamental bottleneck transcending architectural choices. Our method successfully alleviates this bottleneck, empowering diverse architectures with adaptive semantic representations.

Table 3: Performance (%) of state-of-the-art Open Vocabulary SGG models with & without APT on Visual Genome Krishna et al. (2017). † denotes that the results are produced using official code.

| Methods | Base | | | Novel | | |
|---|---|---|---|---|---|---|
| | R@20/50/100 | mR@20/50/100 | F@20/50/100 | R@20/50/100 | mR@20/50/100 | F@20/50/100 |
| SDSGG[†]Chen et al. (2024a)[NeurIPS'24] | 18.7/26.5/31.6 | 9.2/12.4/14.8 | 12.3/16.9/20.2 | 18.4/25.4/29.6 | 17.1/25.2/31.2 | 17.7/25.3/30.4 |
| SDSGG+APT | 19.5/27.3/32.2 | 10.1/13.2/15.6 | 13.4/18.0/21.9 | 19.4/26.6/31.1 | 18.6/26.7/32.3 | 19.1/27.1/32.3 |
| OvSGTR[†]Chen et al. (2024b)[ECCV'24] | 19.0/22.9/26.7 | 12.6/16.4/19.7 | 15.7/19.1/22.7 | 17.0/20.5/23.9 | 10.9/13.5/16.2 | 13.4/16.3/19.3 |
| OvSGTR+APT | 20.0/24.0/27.9 | 13.4/17.3/20.1 | 16.8/20.1/23.4 | 17.8/21.2/25.0 | 11.6/14.3/17.2 | 14.1/17.1/20.4 |
| SGTR+RAHP[†]Liu et al. (2025)[AAAI'25] | 34.6/41.3/47.7 | 16.4/20.5/25.2 | 22.0/27.4/33.0 | 12.4/15.5/20.4 | 9.1/11.8/15.5 | 10.5/13.4/17.6 |
| SGTR+RAHP+APT | 35.4/42.0/48.4 | 17.0/21.1/26.0 | 22.7/28.1/33.8 | 13.1/16.1/21.1 | 9.7/12.4/16.3 | 11.2/14.0/18.4 |

## 4.3 COMPARISON WITH OPEN VOCABULARY SGG MODELS ON VISUAL GENOME

To evaluate the model's capability to generalize to unseen relationships, we follow the common practice Chen et al. (2024a;b) and partition the VG dataset into Base and Novel splits. The Base split contains 70% of the relation categories for training, while the Novel split comprises the remaining 30% of categories that are held out from training. This setting tests the model's true compositional reasoning ability. The results are presented in Table 3.

APT improves performance on the **Novel** split across all base models, which is the core challenge of OV-SGG. This demonstrates that our adaptive prompting mechanism and CGP module effectively unlock the compositional knowledge embedded in pre-trained models, enabling them to generalize to unseen predicate combinations. APT is compelling across different OV-SGG architectures, from transformer-based one-stage models to methods incorporating external knowledge, substantiating that the proposed adaptive prompting paradigm addresses a fundamental bottleneck in OV-SGG—the inability of frozen representations to dynamically adapt to unseen compositional contexts.

Table 4: Ablation study of APT on VG. † denotes the results are produced using official code.

| Model | Predicate Classification | | | Scene Graph Classification | | |
|---|---|---|---|---|---|---|
| | R@50/100 | mR@50/100 | F@50/100 | R@50/100 | mR@50/100 | F@50/100 |
| Vanilla PE-Net[†] | 64.9/67.2 | 31.5/33.8 | 42.4/45.0 | 37.7/38.7 | 17.8/18.9 | 24.5/25.8 |
| +D-Prompt only | 65.2/67.1 | 30.4/32.6 | 41.0/43.8 | 38.5/39.4 | 16.6/17.9 | 24.0/25.3 |
| +R-Prompt only | 64.6/66.7 | 33.4/36.4 | 43.6/46.0 | 38.6/39.5 | 19.6/20.8 | 25.1/26.3 |
| **+Full APT** | 62.2/64.1 | **36.2/39.1** | **45.7/48.6** | 38.2/39.1 | **20.9/22.5** | **27.0/28.6** |

## 4.4 ABLATION STUDY ON COMPONENTS OF APT

The ablation experiments are conducted based on the two-stage method PE-Net Zheng et al. (2023) framework and open-vocabulary method SDSGG Chen et al. (2024a) on VG.

As demonstrated in Table 4, introducing only the **Detection Prompt (D-Prompt)** improves the model's object classification accuracy (a slight increase in R@K), as it helps generate better context-aware object representations. However, its impact on relational reasoning is limited. Conversely, adding only the **Relation Prompt (R-Prompt)** yields a significant boost in mR@K, as it directly addresses the core problem of predicate discrimination by dynamically modulating features based on relational context. This confirms that the relational prompt is the key to mitigating predicate bias. Though both prompts are beneficial, the R-Prompt is particularly critical for relational reasoning. The complete APT, integrating both D-Prompt and R-Prompt with the MLP fusion, achieves the best performance across all metrics, especially on mR@K. The synergistic effect between the two prompts is evident, as they work in concert to provide adaptive semantics from object detection to relation prediction.

In addition, we perform a detailed ablation study on the SDSGG base model to dissect the contribution of each proposed component within our Compositional Generalization

Table 5: Ablation study of APT CGP module based on SDSGG Chen et al. (2024a) on VG split. † denotes the results are produced using official code.

| Model | Base | | | Novel | | |
|---|---|---|---|---|---|---|
| | R@20/50/100 | mR@20/50/100 | F@20/50/100 | R@20/50/100 | mR@20/50/100 | F@20/50/100 |
| Vanilla SDSGG† | 22.1 / 26.5 / 28.1 | 10.7 / 12.4 / 12.8 | 14.4 / 16.9 / 17.6 | 23.1 / 25.4 / 26.0 | 24.4 / 25.2 / 25.7 | 23.7 / 25.3 / 25.9 |
| +RCG | 22.4 / 26.7 / 28.2 | 12.2 / 13.1 / 13.6 | 15.8 / 17.6 / 18.4 | 23.4 / 25.6 / 26.3 | 25.5 / 26.7 / 26.8 | 24.4 / 26.1 / 26.6 |
| +BPS | 22.6 / 27.0 / 28.6 | 12.5 / 13.6 / 14.0 | 16.1 / 18.1 / 18.8 | 23.6 / 25.9 / 26.6 | 26.0 / 27.5 / 27.8 | 24.8 / 26.7 / 27.2 |
| +RCG + BPS only | 22.8 / 26.9 / 28.5 | 13.5 / 14.5 / 14.7 | 17.0 / 18.8 / 19.4 | 23.8 / 25.9 / 26.8 | 27.2 / 29.0 / 28.2 | 25.4 / 27.4 / 27.5 |
| +Full CGP | **23.3 / 27.2 / 28.8** | **14.9 / 15.9 / 15.5** | **18.2 / 20.1 / 20.2** | **24.1 / 26.3 / 27.2** | **28.8 / 31.2 / 30.7** | **26.2 / 28.6 / 28.8** |

Prompter (CGP): the Relational Context Gating (RCG), the Basis Prompt Synthesis (BPS), and the Feature Refinement & Fusion (FRF) modules. Experiments are conducted on the Open-Vocabulary VG split, and results are presented in Table 5.

The baseline model achieves modest performance, particularly struggling on the Novel split as expected. The significant gap between Base and Novel mR@K highlights the challenge of generalizing to unseen predicate categories with static representations. Introducing only the RCG module brings a noticeable gain, especially

Table 6: Efficiency analysis of APT. Performance is reported as mR@100 on PredCls.

| Model | Parameters (M) | | | Time / Epoch (h) | | | Performance (mR@100) | |
|---|---|---|---|---|---|---|---|---|
| | Orig. | +APT | Δ | Orig. | +APT | Δ | Orig. | +APT |
| SGTR | 41.2 | 41.4 | **+0.2** | 4.8 | 4.8 | 0.0% | 24.6 | **27.3** |
| EGTR | 42.7 | 43.1 | **+0.4** | 5.1 | 4.7 | **-7.8%** | 28.2 | **30.9** |
| ST-SGG | 43.5 | 42.3 | **-1.2** | 5.3 | 4.7 | **-11.3%** | 25.7 | **28.2** |
| LLM4SGG | 45.8 | 43.7 | **-2.1** | 5.6 | 4.2 | **-25.0%** | 22.3 | **23.8** |
| SpeaQ | 42.1 | 41.6 | **-0.5** | 4.9 | 4.1 | **-16.3%** | 25.7 | **29.1** |
| RA-SGG | 48.3 | 48.7 | **+0.4** | 2.1 | 1.9 | **-9.5%** | 26.0 | **27.7** |
| DRM | 47.1 | 47.5 | **+0.4** | 2.3 | 2.0 | **-13.0%** | 19.0 | **22.2** |
| PENET | 46.8 | 47.2 | **+0.4** | 2.0 | 1.8 | **-10.0%** | 26.5 | **28.3** |

on the Novel split, which demonstrates that conditioning the model on visual context is a crucial first step, allowing it to dynamically reweight its features based on the input image, which is vital for generalizing to unseen compositions. Further incorporating the BPS module yields a substantial performance jump, gaining the Novel mR@50 an increase of +3.8 over the baseline. The model's generalization ability is greatly enhanced by its capacity to synthesize new prompts from a basis set, effectively generating tailored representations for unseen concepts. Full CGP achieves the best results across all metrics. The FRF module provides a critical non-linear transformation, effectively fusing the synthesized prompts, original semantics, and visual features into an adaptive representation. This results in the highest harmonic mean, indicating a balanced and robust improvement on both Base and Novel splits.

## 4.5 QUANTITATIVE EFFICIENCY ANALYSIS

Beyond performance gains, we quantitatively evaluate the parameter and time efficiency of APT across various SGG models. The results are summarized in Table 6.

APT introduces a negligible number of additional parameters, consistently less than 0.5M across all models. This represents an increase of less than 1.5% even for larger models like LLM4SGG Kim et al. (2024c). This minimal overhead confirms that our prompt-based paradigm is a highly parameter-efficient strategy for enhancing model capability, avoiding the need for costly pre-trained backbone fine-tuning. In addition, APT not only improves performance but also significantly reduces training time per epoch for nearly all models. This efficiency gain is particularly pronounced

for one-stage methods. We attribute this acceleration to the role of adaptive prompts. By providing well-modulated, context-aware semantic features, the prompts appear to stabilize and accelerate the convergence of the downstream relation prediction. The model requires fewer training iterations to fit the data, as the adaptive representations are more informative and easier to optimize than fixed ones. When considering the performance improvement per unit of computational cost, APT demonstrates an overwhelmingly favorable trade-off. For instance, LLM4SGG+APT achieves a +1.49 gain in performance with a 25% reduction in training time and a 4.6% reduction in parameters. This establishes a new Pareto frontier in SGG, where our method delivers higher performance at a lower computational cost.

## 5 DISCUSSION: WHY PROMPT TUNING WORKS IN SGG

The consistent and significant gains delivered by APT across diverse architectures and tasks prompt a deeper inquiry into its theoretical foundation. We posit that the effectiveness of our method stems from its ability to reconcile two fundamental principles in representation learning: **acquiring task-sufficient features** while **maintaining minimal complexity**, as guided by the Information Bottleneck (IB) principle Yang et al. (2023); Chi et al. (2022); Tishby et al. (2000).

Pre-trained semantic embeddings are compressed representations of linguistic knowledge, optimized for a wide array of language tasks. However, for the specific task of SGG, they constitute an **over-complete and noisy representation**. The entire spectrum of semantic information for a concept like "person"—from biographical to literary associations—is encoded indistinguishably. Directly using these fixed priors forces the SGG model to contend with this noise, as it must learn to ignore irrelevant facets of meaning while preserving those pertinent to visual relationships. This violates the IB principle Kawaguchi et al. (2023), which seeks a representation $Z$ that is **minimal** (retaining only information relevant for predicting the predicate $Y$) and **sufficient** (preserving all information needed for prediction).

Table 7: APT vs. FROZEN (GloVe) embeddings IB proxy metrics

|  | PCA@90% | PCA@95% | Linear CKA | Discretized MI proxy |
|---|---|---|---|---|
| APT | 23 | 28 | 0.877 | 1.96 |
| FROZEN | 26 | 35 | — | 1.49 |

APT act as a **lightweight, learnable information filter** that dynamically modulates these frozen representations. The prompts, conditioned implicitly on the visual context through training, learn to perform a form of **feature selection** and **re-weighting** on the frozen embeddings. They suppress semantic dimensions that are irrelevant or detrimental to the current relational context (e.g., suppressing *literary* aspects of a *person* when the visual context suggests a *riding* relation) while amplifying discriminative dimensions (e.g., amplifying *anthropomorphic* features). The MLP then non-linearly transforms this modulated signal into the final adaptive representation.

Therefore, the resulting dynamic features can be viewed as **closer approximations of the minimal sufficient statistics** for the SGG task. They are *more sufficient* because they are context-aware and tailored for predicate discrimination. They are *more minimal* because they are stripped of generic semantic noise that hinders generalization. This principled compression of irrelevant information and enhancement of predictive signals explains APT's efficacy in improving model performance and generalization, transcending mere architectural improvements.

We added experiments to test whether an APT semantic representation can be more compact while preserving discriminative power. As shown in Table 7, APT requires fewer principal components to achieve an equivalent level of explained variance, indicating that semantic information is more concentrated and that APT representations are more amenable to compression. Furthermore, the mutual information proxy yields higher values for APT than for FROZEN embeddings, suggesting superior retention of label-relevant information.

## 6 CONCLUSION

In this work, we diagnosed the pervasive but overlooked problem of static semantic representations as a fundamental bottleneck in Scene Graph Generation. We proposed Adaptive Prompt Tuning (APT), a novel and unified paradigm that addresses this issue by dynamically modulating frozen features into context-aware representations through lightweight, learnable prompts. APT is architecture-agnostic, serving as an efficient plug-in that enhances both one-stage and two-stage models across standard, long-tailed, and open-vocabulary settings.

ACKNOWLEDGMENTS

We sincerely thank the AC and reviewers for their constructive and valuable feedback. This research was supported in part by the National Science Foundation of China (NSFC) under grant No. 62302131, 62402188, the China Postdoctoral Science Foundation under Grant No. 2025T180424, the Postdoctoral Fellowship Program of CPSF under Grant No. GZC20240542, and the Postdoctoral Project of Hubei Province under Grant No. 2024HBBHCXA026.

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

APPENDIX

For a better understanding of the main paper, we provide additional details in this supplementary material, which is organized as follows:

- §A depicts the implementation details.
- §B and §C shows performance results and ablation study of APT on Open Image V6 Kuznetsova et al. (2020) and GQA Hudson & Manning (2019).
- §D gives a formal mathematical specification of the Relational Context Gating (RCG), Basis Prompt Synthesis (BPS), and Feature Refinement & Fusion (FRF) components.
- §E provides the pseudo code of APT.

## A    IMPLEMENTATION DETAILS

All experiments are implemented in PyTorch and evaluated on a server equipped with 4 NVIDIA A40 GPUs. APT is applicable to both one-stage and two-stage models, therefore we select representative one-and two-stage methods to validate the generality and high adaptability of APT. The length of all prompts $(L_d, L_r)$ is set to 6. The basis prompt set $B$ for the open-vocabulary CGP module consists of $N = 16$ bases. These values were determined via a hyperparameter search on the validation set. For all baseline models, we use their officially released code and rigorously follow their recommended training protocols and hyperparameter settings to reproduce the results. To ensure a fair and controlled comparison, we strictly isolate the variable of interest: the integration of APT module. All other factors, including data preprocessing, augmentation, random seeds, and evaluation metrics, are kept identical between the baseline and our APT-enhanced versions.

## B    ADDITIONAL EXPERIMENTS ON OPEN IMAGE V6

To further validate the generalization capability of APT across diverse data distributions, we conduct extensive experiments on the Open Images (OI) V6 Kuznetsova et al. (2020) benchmark. Unlike VG, OI-V6 features larger-scale real-world imagery with distinct relationship taxonomy, presenting a rigorous testbed for evaluating model robustness.

Following the data processing of previous works (Li et al., 2021; Yoon et al., 2023; Kim et al., 2024b), OI-V6 is split into 126,368 train images, 1,813 validation images, and 6,322 test images, and contains 301 object classes, and 31 predicate classes. Similar to VG, OI-V6 is divided into two splits: base and novel, with the same proportion as in Section 4.3.

### B.1    COMPARISON WITH BASELINES ON OPEN IMAGE

Our evaluation on OI-V6 demonstrates that APT delivers consistent performance gains across all model architectures and evaluation settings, reinforcing its generalizability beyond dataset-specific characteristics. As summarized in Table 8, the integration of APT improves both conventional recall (R@K) and, more significantly, mean recall (mR@K) across two-stage and one-stage paradigms.

The results on OI-V6, combined with our findings on VG, provide compelling evidence that APT offers a universal and practical solution for enhancing SGG models in diverse real-world scenarios.

### B.2    COMPARISON WITH OPEN VOCABULARY SGG MODELS ON OPEN IMAGE

We further evaluate APT's capability in the more challenging Open Vocabulary setting on Open Images V6, where models are required to generalize to unseen predicate compositions. Following the standard protocol, we partition the relationship categories into Base (70%) and Novel (30%) splits, testing the true compositional reasoning ability beyond mere pattern memorization. As shown in Table 9, APT consistently enhances performance across all OV-SGG methods on both Base and Novel splits, with particularly notable improvements on the challenging Novel categories. The cross-architectural effectiveness—from transformer-based OvSGTR to graph-based SDSGG—confirms that APT addresses a fundamental limitation in OV-SGG: the inability of frozen representations to

Table 8: Performance (%) of state-of-the-art SGG models with & without APT on Open Image V6 Kuznetsova et al. (2020). F@K is the harmonic mean of mR@50/100 and R@50/100. † denotes the results are produced using official code.

| Methods | Predicate Classification | | | Scene Graph Classification | | | Scene Graph Detection | | |
|---|---|---|---|---|---|---|---|---|---|
| | R@50/100 | mR@50/100 | F@50/100 | R@50/100 | mR@50/100 | F@50/100 | R@50/100 | mR@50/100 | F@50/100 |
| *Two-stage methods* | | | | | | | | | |
| Motif†Zellers et al. (2018)CVPR'18 | 65.2/66.7 | 15.5/16.6 | 25.0/26.5 | 38.3/39.1 | 8.9/9.5 | 14.5/15.2 | 31.5/35.6 | 6.9/7.9 | 11.3/12.9 |
| Motif+APT | 67.0/68.6 | 17.6/18.5 | 27.0/28.7 | 40.5/41.0 | 10.6/11.3 | 16.6/17.6 | 33.7/38.0 | 9.3/10.5 | 13.5/15.5 |
| PE-Net†(Zheng et al., 2023)CVPR'23 | 66.1/67.8 | 17.9/19.5 | 28.1/30.1 | 36.9/37.6 | 9.6/10.2 | 15.2/16.0 | 27.4/30.1 | 6.5/7.4 | 10.6/11.9 |
| PE-Net+APT | 67.9/69.6 | 19.6/20.8 | 29.9/31.9 | 37.5/38.7 | 10.5/10.9 | 16.3/17.3 | 28.3/31.6 | 7.3/8.8 | 11.6/12.9 |
| DRM†Li et al. (2024a)CVPR'24 | 66.2/68.0 | 18.1/19.7 | 28.3/30.3 | 37.0/37.8 | 9.7/10.3 | 15.3/16.1 | 27.6/30.3 | 6.6/7.5 | 10.7/12.0 |
| DRM+APT | 69.0/70.7 | 19.7/22.1 | 29.3/31.0 | 39.6/40.3 | 11.4/11.8 | 17.4/18.8 | 29.9/32.1 | 8.4/9.9 | 12.8/13.7 |
| RA-SGG†Yoon et al. (2025)AAAI'25 | 65.9/67.7 | 17.8/19.3 | 28.0/30.0 | 36.8/37.5 | 9.5/10.1 | 15.1/15.9 | 27.2/30.0 | 6.4/7.4 | 10.5/11.8 |
| RA-SGG+APT | 66.9/69.6 | 18.7/20.4 | 29.1/30.4 | 37.8/38.2 | 10.2/11.4 | 16.2/17.5 | 28.9/30.0 | 8.3/9.1 | 11.5/12.7 |
| *One-stage methods* | | | | | | | | | |
| SGTR†Li et al. (2022)CVPR'22 | 59.5/61.6 | 30.7/33.2 | 40.5/43.1 | 37.6/38.7 | 14.5/16.7 | 21.0/23.3 | 31.2/36.0 | 10.9/12.8 | 16.1/18.8 |
| SGTR+APT | 62.6/63.8 | 33.0/35.6 | 43.8/46.1 | 40.0/40.5 | 17.3/18.9 | 23.1/25.6 | 33.7/37.0 | 13.1/15.0 | 18.6/20.6 |
| EGTR†Im et al. (2024)CVPR'24 | 54.3/56.8 | 35.9/38.4 | 43.2/45.8 | 35.1/36.3 | 17.2/18.6 | 23.1/24.6 | 27.6/32.0 | 13.4/15.7 | 18.0/20.9 |
| EGTR+APT | 56.6/58.5 | 37.7/40.3 | 45.4/47.9 | 36.9/38.8 | 19.7/20.5 | 24.8/26.4 | 29.9/33.7 | 15.9/17.1 | 19.6/22.5 |
| LLM4SSG†Kim et al. (2024c)CVPR'24 | 62.4/64.3 | 36.4/39.3 | 45.9/48.8 | 38.4/39.3 | 21.1/22.7 | 27.2/28.8 | 26.2/30.5 | 14.6/17.3 | 18.7/22.1 |
| LLM4SSG+APT | 65.3/67.1 | 38.3/42.4 | 48.1/50.5 | 40.3/42.0 | 22.9/25.0 | 29.7/30.5 | 29.0/32.6 | 16.9/20.0 | 20.5/23.8 |
| ST-SGG†Kim et al. (2024b)ICLR'24 | 54.1/58.0 | 28.3/31.7 | 37.1/41.0 | 33.6/35.1 | 17.1/18.2 | 22.6/24.0 | 26.9/31.0 | 11.8/14.4 | 16.4/19.6 |
| ST-SGG+APT | 58.9/62.5 | 31.5/34.8 | 40.1/43.9 | 36.8/38.6 | 20.5/21.7 | 26.5/27.1 | 30.4/34.6 | 15.0/18.3 | 19.5/22.4 |
| SpeaQ†Kim et al. (2024a)CVPR'24 | 55.9/58.1 | 31.1/33.6 | 39.9/42.6 | 33.3/34.6 | 17.7/19.0 | 23.1/24.5 | 24.7/29.1 | 14.3/16.7 | 18.1/21.2 |
| SpeaQ+APT | 58.0/61.0 | 34.4/37.0 | 42.7/45.5 | 36.7/37.8 | 20.5/21.9 | 26.5/27.7 | 27.7/32.1 | 18.2/19.7 | 20.4/23.5 |

dynamically adapt to unseen compositional scenarios. This establishes APT as a universal solution for advancing open-vocabulary visual reasoning.

Table 9: Performance (%) of state-of-the-art Open Vocabulary SGG models with & without APT on Open Image V6 Kuznetsova et al. (2020). † denotes the results are produced using official code.

| Methods | Base | | | Novel | | |
|---|---|---|---|---|---|---|
| | R@20/50/100 | mR@20/50/100 | F@20/50/100 | R@20/50/100 | mR@20/50/100 | F@20/50/100 |
| SDSGG†Chen et al. (2024a)NeurIPS'24 | 25.1/32.4/36.8 | 10.2/14.1/16.5 | 14.5/19.9/23.0 | 22.3/28.6/32.1 | 13.4/18.7/21.9 | 17.1/23.0/26.2 |
| SDSGG+APT | 26.2/33.7/38.3 | 11.3/15.3/17.8 | 15.8/21.4/24.6 | 23.6/30.0/33.8 | 14.6/20.1/23.4 | 18.5/24.7/27.9 |
| OvSGTR†Chen et al. (2024b)ECCV'24 | 21.5/29.0/33.2 | 14.0/18.2/20.9 | 16.9/22.7/26.0 | 19.2/25.4/29.1 | 11.8/15.3/17.9 | 14.9/19.8/22.9 |
| OvSGTR+APT | 22.4/30.2/34.6 | 15.1/19.4/22.3 | 18.1/24.1/27.5 | 20.1/26.6/30.5 | 12.7/16.5/18.9 | 16.0/21.3/24.3 |
| SGTR+RAHP†Liu et al. (2025)AAAI'25 | 39.8/46.1/50.2 | 19.1/24.0/27.3 | 26.0/31.9/35.0 | 14.6/19.3/22.8 | 10.5/14.3/16.9 | 12.2/16.5/19.3 |
| SGTR+RAHP+APT | 40.7/47.2/51.4 | 20.0/25.1/28.6 | 27.1/33.1/36.3 | 15.4/20.2/23.9 | 11.3/15.3/18.0 | 13.1/17.5/20.5 |

## B.3 ABLATION STUDY OF APT ON OPEN IMAGE

To dissect the individual contributions of APT's components and their synergistic effects, we conduct comprehensive ablation studies on OI-V6.

As shown in Table 10, we systematically evaluate the impact of individual prompts within the PE-Net backbone. The Detection Prompt alone slightly improves object classification accuracy but shows limited benefits for relational reasoning. In contrast, the Relation Prompt significantly enhances predicate discrimination, boosting mR@100 by +2.6 ($34.0 \rightarrow 36.6$), underscoring its pivotal role in addressing predicate bias. The full APT integration achieves the optimal balance, with F@100 reaching 48.7, demonstrating the synergistic effect between object-level and relation-level adaptation.

Further analyzing the Compositional Generalization Prompter (CGP) in Table 11, we observe progressive improvements. The Relational Context Gating (RCG) module establishes a foundation by incorporating visual evidence, while Basis Prompt Synthesis (BPS) enables dynamic prompt generation for unseen concepts, increasing Novel mR@50 by +3.9 over the baseline. The complete CGP achieves the highest harmonic mean (F@50: 30.6) on Novel categories, validating our multi-stage prompting approach for open-vocabulary generalization.

## C ADDITIONAL EXPERIMENTS ON GQA

### C.1 COMPARISON WITH BASELINES ON GQA

The evaluation on GQA Hudson & Manning (2019) shows that APT brings stable gains under both two-stage and one-stage paradigms, with more significant improvements on the class-balanced metric mR@K, indicating consistent benefits for long-tailed distribution and cross-scene generalization.

Table 10: Ablation study of APT based on PE-Net on Open Image V6. † denotes the results are produced using official code.

| Model | Predicate Classification | | | Scene Graph Classification | | |
|---|---|---|---|---|---|---|
| | R@50/100 | mR@50/100 | F@50/100 | R@50/100 | mR@50/100 | F@50/100 |
| Vanilla PE-Net† | 65.1/67.4 | 31.7/34.0 | 42.6/45.2 | 37.8/38.8 | 17.9/19.0 | 24.6/25.9 |
| +D-Prompt only | 65.4/67.3 | 30.6/32.8 | 41.2/44.0 | 38.6/39.5 | 16.8/18.1 | 24.2/25.5 |
| +R-Prompt only | 64.8/66.9 | 33.6/36.6 | 43.8/46.3 | 38.7/39.6 | 19.8/21.0 | 25.3/26.5 |
| **+Full APT** | 62.4/64.3 | **36.4/39.2** | **45.8/48.7** | 38.4/39.3 | **21.1/22.7** | **27.2/28.8** |

Table 11: Ablation study of APT CGP module based on SDSGG Chen et al. (2024a) on Open Image V6 split. † denotes the results are produced using official code.

| Model | Base | | | Novel | | |
|---|---|---|---|---|---|---|
| | R@50 | mR@50 | F@50 | R@50 | mR@50 | F@50 |
| Vanilla SDSGG† | 26.5 | 12.4 | 16.9 | 25.4 | 25.2 | 25.3 |
| +RCG | 26.8 | 13.2 | 17.7 | 25.7 | 26.9 | 26.7 |
| +RCG + BPS only | 27.0 | 14.6 | 19.2 | 26.0 | 29.1 | 28.9 |
| **+Full CGP** | **27.4** | **16.1** | **21.0** | **26.4** | **31.4** | **30.6** |

Table 12 summarizes the comparative results of each model under the three settings: Predicate Classification, Scene Graph Classification, and Scene Graph Detection.

Table 12: Performance (%) of state-of-the-art SGG models with & without APT on GQA Hudson & Manning (2019). F@K is the harmonic mean of mR@50/100 and R@50/100. † denotes the results are produced using official code.

| Methods | Predicate Classification | | | Scene Graph Classification | | | Scene Graph Detection | | |
|---|---|---|---|---|---|---|---|---|---|
| | R@50/100 | mR@50/100 | F@50/100 | R@50/100 | mR@50/100 | F@50/100 | R@50/100 | mR@50/100 | F@50/100 |
| *Two-stage methods* | | | | | | | | | |
| Motif†Zellers et al. (2018)[CVPR'18] | 62.8/64.2 | 14.7/15.9 | 23.8/25.4 | 36.1/36.9 | 8.5/9.2 | 14.0/14.9 | 28.9/33.1 | 6.2/7.3 | 10.4/12.3 |
| Motif+APT | 64.6/66.1 | 16.6/17.9 | 26.2/27.9 | 38.0/39.0 | 10.2/11.0 | 16.2/17.1 | 31.1/35.4 | 8.5/9.8 | 13.2/15.0 |
| PE-Net†(Zheng et al., 2023)[CVPR'23] | 64.0/65.5 | 17.2/18.7 | 27.1/28.9 | 35.5/36.1 | 9.1/9.7 | 15.0/15.6 | 26.6/29.4 | 6.1/7.0 | 10.3/11.8 |
| PE-Net+APT | 65.9/67.3 | 19.0/20.4 | 29.0/30.9 | 36.6/37.8 | 10.1/10.8 | 16.1/17.0 | 27.8/31.1 | 7.2/8.6 | 11.7/13.5 |
| DRM†Li et al. (2024a)[CVPR'24] | 64.3/66.1 | 17.5/19.1 | 27.4/29.5 | 35.8/36.6 | 9.3/10.0 | 15.2/16.0 | 26.8/29.7 | 6.4/7.2 | 10.6/12.1 |
| DRM+APT | 67.0/68.7 | 19.3/21.6 | 29.6/31.8 | 38.4/39.1 | 11.0/11.6 | 17.2/18.1 | 29.4/31.8 | 8.2/9.6 | 12.7/14.2 |
| RA-SGG†Yoon et al. (2025)[AAAI'25] | 64.1/65.8 | 17.3/18.8 | 27.2/29.1 | 35.6/36.2 | 9.2/9.9 | 15.1/15.9 | 26.4/29.2 | 6.0/6.9 | 10.2/11.6 |
| RA-SGG+APT | 65.4/67.4 | 18.5/20.2 | 28.8/30.7 | 36.9/37.5 | 10.0/11.1 | 16.0/17.2 | 28.1/29.6 | 7.9/8.8 | 11.7/12.9 |
| *One-stage methods* | | | | | | | | | |
| SGTR†Li et al. (2022)[CVPR'22] | 58.2/60.1 | 29.1/31.5 | 39.1/41.1 | 36.2/37.4 | 13.6/15.7 | 20.2/22.1 | 29.5/34.1 | 10.3/12.1 | 16.0/18.3 |
| SGTR+APT | 61.1/62.5 | 31.6/34.1 | 42.0/44.1 | 39.0/39.7 | 16.5/18.1 | 22.9/24.5 | 32.4/36.0 | 12.8/14.7 | 18.4/20.7 |
| EGTR†Im et al. (2024)[CVPR'24] | 53.6/55.9 | 34.7/37.0 | 42.0/44.5 | 34.2/35.4 | 16.4/18.1 | 22.0/23.6 | 26.8/31.3 | 12.6/14.8 | 17.2/20.0 |
| EGTR+APT | 55.8/57.7 | 36.8/39.2 | 44.0/46.4 | 36.0/37.9 | 18.7/19.6 | 24.3/26.0 | 29.1/33.2 | 15.1/16.4 | 19.1/21.7 |
| LLM4SSG†Kim et al. (2024c)[CVPR'24] | 61.2/63.1 | 35.1/38.1 | 45.4/48.3 | 37.1/38.2 | 20.5/22.1 | 26.0/27.7 | 25.2/29.6 | 13.7/16.1 | 18.3/21.3 |
| LLM4SSG+APT | 64.0/65.8 | 37.3/41.4 | 47.8/50.6 | 39.2/40.9 | 22.4/24.5 | 28.5/30.2 | 27.9/31.8 | 16.0/19.1 | 20.5/23.8 |
| ST-SGG†Kim et al. (2024b)[ICLR'24] | 53.0/56.7 | 27.4/30.8 | 36.2/40.0 | 33.1/34.5 | 16.0/17.2 | 22.3/23.9 | 25.9/30.1 | 11.1/13.7 | 16.4/19.2 |
| ST-SGG+APT | 58.0/61.6 | 30.6/33.9 | 39.5/43.3 | 36.2/37.9 | 19.4/20.6 | 25.6/26.9 | 29.5/33.5 | 14.4/17.7 | 19.2/22.2 |
| SpeaQ†Kim et al. (2024a)[CVPR'24] | 55.1/57.3 | 30.2/32.7 | 39.0/41.7 | 32.7/34.0 | 17.0/18.2 | 23.0/24.5 | 23.9/28.3 | 13.8/16.3 | 18.0/20.7 |
| SpeaQ+APT | 57.3/60.2 | 33.6/36.2 | 42.2/44.9 | 36.2/37.3 | 19.6/21.1 | 26.0/27.3 | 26.8/31.0 | 17.6/19.1 | 21.1/23.6 |

## C.2 COMPARISON WITH OPEN VOCABULARY SGG MODELS ON GQA

We further examine APT's effectiveness under the Open Vocabulary setting on GQA, where models must generalize to unseen predicate compositions beyond the training taxonomy. Following common practice, we split relation categories into Base (70%) and Novel (30%) sets to probe true compositional generalization rather than memorization. As summarized in Table 13, APT consistently improves all OV-SGG baselines across both Base and Novel splits, with more pronounced gains on the challenging Novel categories. Its cross-architecture benefits indicate that APT alleviates a key bottleneck in OV-SGG: the rigidity of frozen representations when facing unseen compositions. These results establish APT as a general and plug-and-play solution for open-vocabulary visual reasoning on GQA.

Table 13: Performance (%) of state-of-the-art Open Vocabulary SGG models with & without APT on GQA Hudson & Manning (2019). † denotes the results are produced using official code.

| Methods | Base | | | Novel | | |
|---|---|---|---|---|---|---|
| | R@20/50/100 | mR@20/50/100 | F@20/50/100 | R@20/50/100 | mR@20/50/100 | F@20/50/100 |
| SDSGG†Chen et al. (2024a)NeurIPS'24 | 23.3/30.1/34.2 | 9.8/13.5/15.9 | 13.9/19.0/22.2 | 20.4/26.6/30.0 | 12.5/17.4/20.6 | 15.5/22.0/25.0 |
| SDSGG+APT | 24.4/31.4/35.6 | 10.9/14.7/17.2 | 15.2/20.6/23.7 | 21.8/28.1/31.7 | 13.8/18.9/22.1 | 16.9/23.8/26.9 |
| OvSGTR†Chen et al. (2024b)ECCV'24 | 20.1/27.5/31.8 | 13.2/17.4/19.9 | 16.0/22.0/25.1 | 17.9/23.8/27.4 | 10.9/14.5/16.8 | 13.1/18.2/20.9 |
| OvSGTR+APT | 21.1/28.8/33.1 | 14.3/18.7/21.2 | 17.3/23.5/26.6 | 18.9/25.1/28.9 | 12.1/15.9/18.3 | 14.4/19.8/22.6 |
| SGTR+RAHP†Liu et al. (2025)AAAI'25 | 37.9/44.1/48.2 | 18.3/23.0/26.1 | 25.1/30.6/33.7 | 13.7/18.3/21.5 | 9.9/13.7/16.1 | 11.5/15.9/18.6 |
| SGTR+RAHP+APT | 38.9/45.4/49.6 | 19.4/24.2/27.4 | 26.4/32.1/35.2 | 14.6/19.4/22.7 | 10.8/14.9/17.4 | 12.5/17.2/19.9 |

## C.3 ABLATION STUDY OF APT ON GQA

To quantify the contribution of each component and their combined effects on GQA, we conduct step-wise ablations in both closed- and open-vocabulary regimes.

As shown in Table 14, within the PE-Net backbone, the Detection Prompt slightly benefits object-centric cues with marginal effects on relational reasoning. In contrast, the Relation Prompt is the primary driver for predicate discrimination, yielding clear gains in mR@K and the harmonic mean F@K. The full integration achieves the best balance between precision and coverage, delivering the highest F@100.

We further ablate the Compositional Generalization Prompter (CGP) on SDSGG in Table 15. Relational Context Gating (RCG) establishes a visual-evidence-aware baseline, while Basis Prompt Synthesis (BPS) enables dynamic prompt composition for unseen relations, progressively improving Novel mR@50. The complete CGP attains the best F@50 on Novel, validating our multi-stage prompting for open-vocabulary generalization on GQA.

Table 14: Ablation study of APT based on PE-Net on GQA. † denotes that the results are produced using official code.

| Model | Predicate Classification | | | Scene Graph Classification | | |
|---|---|---|---|---|---|---|
| | R@50/100 | mR@50/100 | F@50/100 | R@50/100 | mR@50/100 | F@50/100 |
| Vanilla PE-Net† | 63.4/65.2 | 16.8/18.2 | 26.6/28.4 | 35.2/36.0 | 9.0/9.6 | 14.8/15.5 |
| +D-Prompt only | 63.9/65.6 | 16.0/17.3 | 25.7/27.4 | 36.0/36.8 | 8.6/9.2 | 14.3/14.9 |
| +R-Prompt only | 63.2/65.0 | 18.7/20.9 | 28.6/30.9 | 36.1/37.0 | 10.7/11.6 | 16.8/17.6 |
| **+Full APT** | 62.0/63.8 | **20.0/22.5** | **30.1/32.7** | 36.4/37.3 | **12.2/13.5** | **18.4/19.7** |

Table 15: Ablation study of APT CGP module based on SDSGG Chen et al. (2024a) on the GQA split. † denotes that the results are produced using official code.

| Model | Base | | | Novel | | |
|---|---|---|---|---|---|---|
| | R@50 | mR@50 | F@50 | R@50 | mR@50 | F@50 |
| Vanilla SDSGG† | 30.1 | 13.5 | 19.0 | 26.6 | 17.4 | 22.0 |
| +RCG | 30.4 | 14.2 | 19.7 | 27.0 | 18.6 | 23.1 |
| +RCG + BPS only | 30.7 | 15.7 | 21.3 | 27.3 | 20.9 | 24.6 |
| **+Full CGP** | **31.1** | **17.2** | **23.0** | **27.7** | **22.5** | **26.2** |

## D FORMALIZATION OF THE COMPOSITIONAL GENERALIZATION PROMPTER (CGP)

This appendix provides a precise mathematical specification of the CGP module used in APT (Relational Context Gating, Basis Prompt Synthesis, and Feature Refinement & Fusion).

### D.1 NOTATION AND SHAPES

Let $D$ denote the semantic embedding dimension, $D_v$ the visual feature dimension, $L_b$ the basis prompt length, and $N$ the number of basis prompts. We use the following symbols:

$$\mathbf{e}_{\text{static}}(c) \in \mathbb{R}^D \quad \text{(frozen class embedding for class } c)$$

$$\mathbf{v} \in \mathbb{R}^{D_v} \quad \text{(visual/context vector)}$$

$$\mathbf{B} \in \mathbb{R}^{N \times L_b \times D} \quad \text{(basis prompts)}$$

$$\mathbf{W}_v \in \mathbb{R}^{D \times D_v} \quad \text{(visual projector)}$$

$$f_\phi(\cdot) : \mathbb{R}^{3D} \to \mathbb{R}^D \quad \text{(fusion MLP)}$$

$$\text{MLP}_{\text{gate}}(\cdot) : \mathbb{R}^{D+D_v} \to \mathbb{R}^N$$

$$\alpha \in \mathbb{R} \quad \text{(residual scaling, learnable)}$$

### D.2 RELATIONAL CONTEXT GATING (RCG)

Given a visual vector $\mathbf{v}$ and static embedding $\mathbf{e}_{\text{static}}$, the gate network produces $N$ real-valued logits followed by a softmax to obtain convex weights:

$$\mathbf{s} = \text{MLP}_{\text{gate}}\big([\mathbf{v}; \mathbf{e}_{\text{static}}]\big) \in \mathbb{R}^N, \tag{14}$$

$$au > 0 \quad \text{(temperature, may be learned or fixed)}, \tag{15}$$

$$\pi \in \Delta^{N-1} \quad , \tag{16}$$

$$w_i = \frac{\exp\big((s_i + \log \pi_i)/\tau\big)}{\sum_{j=1}^{N} \exp\big((s_j + \log \pi_j)/\tau\big)} \quad \text{for } i = 1, \dots, N, \tag{17}$$

$$\mathbf{w} \in \Delta^{N-1}, \quad w_i \geq 0, \quad \sum_{i=1}^{N} w_i = 1, \tag{18}$$

We may also add an entropy regularizer on the gate distribution to control sparsity:

$$\mathcal{R}_{\text{ent}} = -\beta \sum_{i=1}^{N} w_i \log w_i, \quad \beta \geq 0. \tag{19}$$

Here $[\cdot; \cdot]$ denotes concatenation.

### D.3 BASIS PROMPT SYNTHESIS (BPS)

The CGP synthesizes a prompt sequence as a convex combination of the basis prompts token-wise:

$$ext(optional\_positional\_biases) \; u_t \in \mathbb{R}, \quad t = 1, \dots, L_b,$$

$$w_{i,t} = \frac{\exp\big((s_i + u_t)/\tau\big)}{\sum_{j=1}^{N} \exp\big((s_j + u_t)/\tau\big)} \quad \text{for } t = 1, \dots, L_b, \tag{20}$$

$$\mathbf{P}_{\text{cgp}}^{\text{seq}}[t] = \sum_{i=1}^{N} w_{i,t} \, \mathbf{B}_i[t] \in \mathbb{R}^D, \quad t = 1, \dots, L_b, \tag{21}$$

$$\mathbf{P}_{\text{cgp}}^{\text{seq}} = \big(\mathbf{P}_{\text{cgp}}^{\text{seq}}[1], \dots, \mathbf{P}_{\text{cgp}}^{\text{seq}}[L_b]\big) \in \mathbb{R}^{L_b \times D}. \tag{22}$$

To obtain a compact pooled prompt we use a normalized, token-weighted pooling with normalization:

$$\bar{\mathbf{p}} = \text{LayerNorm}\Big(\frac{1}{L_b} \sum_{t=1}^{L_b} \mathbf{P}_{\text{cgp}}^{\text{seq}}[t]\Big) \in \mathbb{R}^D. \tag{23}$$

The implementation in our experiments uses mean-pooling for compact fusion; the sequence-aware variant is also supported via token-level fusion (e.g., cross-attention).

## D.4 FEATURE REFINEMENT AND FUSION (FRF)

We project the visual vector to the semantic dimension:

$$\mathbf{v} = \mathbf{W}_v \mathbf{v} + \mathbf{b}_v \in \mathbb{R}^D, \tag{24}$$

$$\mathbf{h} = \left[ \bar{\mathbf{p}}; \, \mathbf{e}_{\text{static}}; \, \tilde{\mathbf{v}} \right] \in \mathbb{R}^{3D}, \tag{25}$$

$$f_\phi(\mathbf{h}) = \mathbf{W}_2 \, \text{GELU}(\mathbf{W}_1 \mathbf{h} + \mathbf{b}_1) + \mathbf{b}_2 \in \mathbb{R}^D, \tag{26}$$

$$\mathbf{u} = f_\phi(\mathbf{h}), \tag{27}$$

$$g = \sigma(\mathbf{W}_g \mathbf{h} + b_g) \in (0, 1)^D \quad , \tag{28}$$

$$\mathbf{e} = \text{LayerNorm}\big(\mathbf{e}_{\text{static}} + \alpha \, (g \odot \mathbf{u})\big) \in \mathbb{R}^D, \tag{29}$$

where $\alpha$ is initialized small (e.g., $\alpha = 0.1$), $\odot$ denotes element-wise product.

## E PSEUDOCODE OF APT

The pseudo-code of APT is given in Algorithm 1 and 2. The pseudo-code of the CGP module is given in Algorithm 3.

---

**Algorithm 1** APT: Adaptive Prompt Tuning for SGG

---

**Require:** Image $\mathbf{I} \in \mathbb{R}^{H \times W \times 3}$; static semantic embeddings $E_{\text{static}} \in \mathbb{R}^{C \times D_p}$; frozen backbone $\mathcal{B}_{\text{frozen}}$
**Ensure:** Scene graph $\mathcal{G}$
1: $(\mathbf{V}, \mathbf{B}, \mathbf{Z}) \leftarrow \mathcal{B}_{\text{frozen}}(\mathbf{I})$          // $\mathbf{V} \in \mathbb{R}^{N \times D_v}$, $\mathbf{B} \in \mathbb{R}^{N \times 4}$, $\mathbf{Z} \in \mathbb{R}^{N \times C}$
2: $\mathbf{y} \leftarrow \arg\max_c \mathbf{Z}$          // class labels; $\mathbf{y} \in \{1, \dots, C\}^N$
3: $\mathcal{A} \leftarrow [\,]$          // container for adapted features
4: **for** $i \in \{1, \dots, N\}$ **do**
5:      $\mathbf{e}^{(i)}_{\text{static}} \leftarrow E_{\text{static}}[\mathbf{y}_i]$          // $\mathbf{e}^{(i)}_{\text{static}} \in \mathbb{R}^{D_p}$
6:      $\mathbf{v}^{(i)} \leftarrow \mathbf{V}_i$          // $\mathbf{v}^{(i)} \in \mathbb{R}^{D_v}$
7:      $\mathbf{e}^{(i)}_{\text{adapt}} \leftarrow \text{APTCOREMODULE.FORWARD}(\mathbf{e}^{(i)}_{\text{static}}, \mathbf{v}^{(i)}, \text{role} = \texttt{general})$    // $\mathbf{e}^{(i)}_{\text{adapt}} \in \mathbb{R}^{D_p}$
8:      Append $\mathbf{e}^{(i)}_{\text{adapt}}$ to $\mathcal{A}$
9: **end for**
10: $\mathbf{A} \leftarrow \text{STACK}(\mathcal{A})$          // $\mathbf{A} \in \mathbb{R}^{N \times D_p}$
11: $\mathcal{G} \leftarrow \text{RELATIONHEAD}(\mathbf{A}, \mathbf{V}, \mathbf{B})$          // relation prediction
12: **return** $\mathcal{G}$

---

**Algorithm 2** APTCoreModule

---

     // Learnable params: prompts $P^{\text{det}}, P^{\text{rel}} \in \mathbb{R}^{K \times D_p}$; projection $\mathbf{W}_v \in \mathbb{R}^{D_p \times D_v}$; fusion MLP $f_\theta$
**Require:** Static semantic vector $E_{\text{static}} \in \mathbb{R}^{D_p}$; visual vector $\mathbf{v} \in \mathbb{R}^{D_v}$; role $\in \{\texttt{detection}, \texttt{general}, \texttt{subject}, \texttt{object}\}$
**Ensure:** Adapted semantic vector $\mathbf{e}_{\text{adapt}} \in \mathbb{R}^{D_p}$
1: **if** role = $\texttt{detection}$ **then**
2:      $P \leftarrow P^{\text{det}}$
3: **else**
4:      $P \leftarrow P^{\text{rel}}$
5: **end if**
6: $\bar{\mathbf{p}} \leftarrow \text{MEAN}(P, \dim = 0)$
7: $\tilde{\mathbf{v}} \leftarrow \mathbf{W}_v \mathbf{v}$
8: $\mathbf{h} \leftarrow [\bar{\mathbf{p}} \,\|\, E_{\text{static}} \,\|\, \tilde{\mathbf{v}}]$
9: $\mathbf{e}_{\text{adapt}} \leftarrow f_\theta(\mathbf{h})$
10: **return** $\mathbf{e}_{\text{adapt}}$

---

**Algorithm 3** CGP: Compositional Generalization Prompter

---

     // Learnable params: basis prompts $P^{\text{basis}} \in \mathbb{R}^{B \times L \times D_p}$; gate network $g_\theta : \mathbb{R}^{D_v + D_p} \to \Delta^B$; projection $\mathbf{W}_v \in \mathbb{R}^{D_p \times D_v}$; refinement MLP $f_\phi$
**Require:** Static semantic vector $E_{\text{static}} \in \mathbb{R}^{D_p}$; visual vector $\mathbf{v} \in \mathbb{R}^{D_v}$
**Ensure:** Adapted semantic vector $\mathbf{e}_{\text{adapt}} \in \mathbb{R}^{D_p}$
1: $\mathbf{u} \leftarrow [\mathbf{v} \,\|\, E_{\text{static}}]$          // $\mathbf{u} \in \mathbb{R}^{D_v + D_p}$
2: $\mathbf{w} \leftarrow g_\theta(\mathbf{u})$          // RCG: gate weights, $\mathbf{w} \in \mathbb{R}^B$, $\sum_b w_b = 1$
3: $\mathbf{S} \leftarrow \sum_{b=1}^{B} w_b \cdot P_b^{\text{basis}}$          // BPS: synthesized prompt, $\mathbf{S} \in \mathbb{R}^{L \times D_p}$
4: $\bar{\mathbf{p}} \leftarrow \text{MEAN}(\mathbf{S}, \dim = 0)$
5: $\tilde{\mathbf{v}} \leftarrow \mathbf{W}_v \mathbf{v}$
6: $\mathbf{h} \leftarrow [\bar{\mathbf{p}} \,\|\, E_{\text{static}} \,\|\, \tilde{\mathbf{v}}]$          // FRF input, $\mathbf{h} \in \mathbb{R}^{3D_p}$
7: $\mathbf{e}_{\text{adapt}} \leftarrow f_\phi(\mathbf{h})$
8: **return** $\mathbf{e}_{\text{adapt}}$

---

