# OpenReview forum: "APT: Towards Universal Scene Graph Generation via Plug-in Adaptive Prompt Tuning"
_ICLR.cc/2026/Conference — ICLR 2026 Poster_

### Official Review · Reviewer_m1jU · 2025-10-31

**Soundness:** 3
**Presentation:** 2
**Contribution:** 2
**Rating:** 6
**Confidence:** 3

**Summary:**

This paper points out that the core bottleneck of SGG is the mismatch between pre-trained semantic priors and visual relationship dynamic contexts, rather than a one-stage/two-stage dispute. Propose an APT lightweight plugin that converts frozen features into context sensitive representations, with 0.5M parameters and a training acceleration of 7.8% -25% plug and play. With the highest score increase of+6.0 on PredCls, SOTA is refreshed with<1.5% overhead, providing SGG with a unified, efficient, and scalable new paradigm.

**Strengths:**

1. The paper is written clearly and organized, especially in the introduction section, where the author provides a detailed explanation of the problem of frozen features using three figures.

2. The APT model not only achieves a significant improvement in accuracy relative to SOTA, but also ensures a limited increase in parameter count and saves training time.

3. The relevant work has been thoroughly sorted out, and the source code of the paper has been provided.

**Weaknesses:**

1. The main objective of this work is to address the shortcomings of the one-stage and two-stage methods, but these shortcomings are not unique to SGG tasks, and the designed method can also be applied to other tasks. This will bring two sub-issues: 1) Have you tried using it on other tasks; 2) The analysis of the specificity of SGG tasks in this work needs to be further explored.

2. The ablation study is comprehensive only under the top-50 setting, while it is incomplete under the top-20/100 setting. In the experimental section, ablation research is the core experiment, and it is strongly recommended to supplement it completely. In addition, it is recommended to try removing the one-stage, two-stage, and Open Vocabulary parts of APT-enhanced methods in sequence to eliminate random factors caused by model combinations.

3. Partial modifications needed: 1) Spelling error in line 045 for "Fiugre"; 2) Some images have font sizes that are too small and do not match the main text.

4. One-stage methods lack the latest methods for 2025.

**Questions:**

See [Weaknesses].

---

> ### Author Response · Authors · 2025-11-21
>
> (**W** refers to **Weakness**; The modifications have been highlighted in **purple in the revised manuscript**.)
>
> -   **Response to W1**
>
> We thank the reviewer for the thoughtful comments.
>
> **Clarification first**: the main objective of this work is not to merely “address shortcomings of one-stage and two-stage methods,” but to diagnose and remove a more fundamental representational bottleneck in SGG—the rigidity of frozen semantic priors coming from pre-trained language models. These static embeddings, while broadly useful elsewhere, are misaligned with SGG’s context-sensitive, role-asymmetric, and relation-centric nature. Our contribution is a universal, lightweight, plug-in representation paradigm that adaptively modulates frozen language features according to visual context and subject/object roles, and works across both one- and two-stage SGG and open-vocabulary settings.
>
> (1) **On applying APT to other tasks**. APT’s idea—turning frozen language features into context-aware representations without backprop through the backbone—is in principle transferable. However, this paper scopes its claims to SGG, and we have already demonstrated broad generality within SGG: across paradigms (one-/two-stage, open-vocabulary) and datasets (VG, OIv6, GQA), with consistent improvements in mR/F under both closed- and open-vocabulary settings. To keep the work focused, we did not include full cross-task evaluations, refraining from over-claiming beyond SGG.
>
> (2) **On SGG specificity and deeper analysis**. The frozen-semantics bottleneck is especially acute in SGG due to:
>
> The same noun (e.g., person) should carry distinct semantics in subject vs object roles; frozen priors cannot express this, whereas APT explicitly models role-conditioned prompts.
>
> Fine-grained relations (e.g., “on” vs “standing on” vs “walking on”) hinge on visual context; our t-SNE analyses (Figure 2) show static features collapse instances whereas visual-context features can separate.
>
> Frozen priors amplify head-class bias and hurt long-tail predicates—our mR/F gains and open-vocabulary improvements highlight APT’s adaptive benefits here.
>
> Simply replacing GloVe with stronger frozen models (BERT/CLIP-text) reveals richer substructures but still leaves them misaligned with SGG’s relational granularity; APT addresses semantic adaptivity rather than adding yet another architecture.
>
> While APT is architecturally plug-and-play and likely transferable, our claims and evidence are focused on SGG, where the frozen-semantics rigidity is uniquely detrimental due to role asymmetry, context dependence, and triadic relational reasoning. We appreciate the suggestion and will add implementation guidance for other tasks and deeper SGG-specific analyses in the revised version without expanding the paper’s scope beyond SGG.
>
> ---
>
> -   **Response to W2**
>
> We have updated the revised manuscript to include the ablation‑study results for the 20/100 settings and, per your suggestion, added the requested ablation experiments. Details are presented in **Table 5 in the revised version**.
>
> The results follow the pattern before. We also added new ablation where we introduce BPS only, gaining the Novel mR@20/50/100 an increase of +1.6/+2.3/+2.1 over the baseline. The model's generalization ability is greatly enhanced by BPS’s capacity to synthesize new prompts from a basis set, effectively generating tailored representations for unseen concepts.
>
> ---
>
> -   **Response to W3**
>
> Thanks for the comments. We have corrected the typos and updated the images as suggested.
>
> ---
>
> -   **Response to W4**
>
> We evaluated the new baseline HQSG [1] (CVPR 2025), with additional experimental results reported in **Table 2 of the revised manuscript**. The results indicate that APT maintains a clear advantage on HQSG. The integration of APT consistently enhances the performance of all base models across the three canonical scene graph generation tasks: PredCls, SGCls, and SGDet.
>
>
>
> [1] Hybrid reciprocal transformer with triplet feature alignment for scene graph generation. In CVPR, pp.8953-8963, 2025.

---

> ### Author Response · Authors · 2025-11-26
>
> Dear Reviewer m1jU
>
> As we near the end of the author-reviewer discussion phase, we would like to sincerely thank you again for your time and valuable feedback. If there are any remaining questions or points you would like us to clarify, please feel free to let us know. We’re here to support the discussion as best we can.
>
> Best regards,
>
> The Authors

---

### Official Review · Reviewer_mifr · 2025-10-31

**Soundness:** 2
**Presentation:** 2
**Contribution:** 2
**Rating:** 4
**Confidence:** 3

**Summary:**

The paper proposes APT, a plug-in module for Scene Graph Generation (SGG) that keeps language backbones frozen while learning lightweight prompts to modulate object/predicate semantic embeddings using visual context. Concretely, APT introduces (i) a Detection Prompt and Relation Prompt fused with static word vectors via small MLPs, and (ii) a Compositional Generalization Prompter (CGP) with context gating, basis‐prompt synthesis, and feature refinement to improve open-vocabulary generalization. The authors motivate APT by arguing that frozen embeddings (GloVe/BERT/CLIP-text) are rigid and misaligned with context-dependent visual relations, illustrated via t-SNE plots and discussion of the one- vs two-stage SGG divide. On VG (with appendix results on OI-V6/GQA), they report consistent gains in mR/F across several backbones and claim <0.5M extra parameters and 7.8%–25% training-time reduction per epoch.

**Strengths:**

1. Clear problem framing. The paper articulates a real pain point in SGG—static semantic priors—and gives intuitive illustrations (Figures 2–3).
2. The idea of drop-in prompts for both one- and two-stage SGG and OV-SGG is straightforward and potentially practical.
3. Compute friendly. Claims of small parameter overhead and faster epochs are attractive for the community.

**Weaknesses:**

1. The core idea (i.e., learn prompts on frozen backbones) has strong prior art: VPT (vision prompts), MaPLe (joint vision+text prompts for CLIP), DualPrompt (complementary/global-vs-expert prompts), and adapter-style PEFT (CLIP-Adapter / Tip-Adapter). APT reads like a tailored application of these ideas to SGG, but the paper does not articulate a new principle beyond that specialization. A side-by-side comparison (same budgets) against these methods is missing.
2. Motivation evidence is mostly qualitative. The "frozen semantics are rigid" claim is intuitive, but support rests on t-SNE/intuition. Provide quantitative diagnostics that isolate the bottleneck (e.g., probing tasks for role asymmetry, controlled swaps of subject/object labels, ablate text vs. vision prompts separately).
3. The information-bottleneck framing is descriptive; provide any measurable prediction (e.g., MI/CKA proxies showing APT features are "more minimal yet sufficient") that the experiments then corroborate.

**Questions:**

1. In Table 5, Why do some+APT variants show fewer parameters than their baselines? Provide a breakdown and add inference latency/memory/FLOPs.

---

> ### Author Response · Authors · 2025-11-21
>
> (**W** refers to **Weakness**; **Q** refers to **Question**; The modifications have been highlighted in **purple in the revised manuscript**.)
>
> -   **Response to W1**
>
> Thank you for pointing out the relevant prior work. We agree these prior methods share the general prompt paradigm. However, those methods were primarily developed for general vision-language tasks, and they operate at a different modeling level than SGG.  APT's contribution is not merely reusing existing prompt techniques; rather, we redefine and implement prompts for structure-aware, role-conditioned relational reasoning at the triplet level (subject–predicate–object). The differences are substantive, both in motivation and design.
>
> **On task difference**:
>
> Prior methods such as VPT, MaPLe, DualPrompt, and CLIP-Adapter/Tip-Adapter mainly target classification or modality alignment tasks (e.g., image classification, zero-shot retrieval, vision–language alignment). Their methods are typically “modality-level” or “token-level” modulations (global token modulation, prefix tuning, or inserted adapters) intended to alter or augment single-entity representations.
>
> SGG is a structured prediction problem: it requires predicting relational triplets where predicates depend on two entities’ roles and on fine-grained visual interactions and context. Hence, SGG demands prompts that support role-conditioned and relation-aware modulation, rather than a single global prefix or adapter that treats each entity independently.
>
> Concretely, instead of only serving as embedding prefixes, APT synthesizes prompts conditioned on subject/object roles and visual context, and it composes novel prompts from a learned basis set to generalize to unseen compositions. This basis-prompt composition is different from token-level prefixes or simple adapter insertions: APT operates at the triplet level and models semantic interactions between entities rather than only adjusting single-token embeddings.
>
> **On Motivation difference:**
>
> Existing work primarily aims to retain pre-trained model knowledge while enabling low-cost adaptation for classification or alignment. In contrast, our central insight is that frozen semantic priors cause a representational collapse in SGG (e.g., all “person” instances share the same static embedding regardless of relational role), which is a cross-architecture bottleneck for relation discrimination. Therefore, our objective is to design prompts that explicitly decouple entity identity from relation role and to inject role- and context-specific information into semantic representations—a principled objective tailored to structured relational reasoning.
>
> In SGG,  prompt-tuning act as selective filters that suppress irrelevant semantic dimensions and amplify relation-relevant dimensions under role conditioning. This targeted modulation better suits the SGG requirement for role-conditioned discrimination while retaining the benefits of frozen pretraining.
>
> ---
> -   **Response to W2**
>
> We thank the reviewers for the suggestion to provide quantitative experiments to evaluate whether frozen semantic embeddings are a representational bottleneck.
>
> *Table 1. Semantic embeddings (GloVe): within‑class distances*
>
> | Subject class | mean_glove_within | std    | n    |
> | ------------- | ----------------- | ------ | ---- |
> | bears         | 0.1033            | 0.0000 | 2    |
> | veil          | 0.1864            | 0.0000 | 2    |
> | painting      | 0.0924            | 0.0678 | 211  |
> | plants        | 0.1602            | 0.0491 | 106  |
> | pedestrian    | 0.1352            | 0.0695 | 24   |
>
> Many subject classes show mean within‑class distances in the roughly 0.05–0.18 range. In the simulated 50‑D embedding space, instances of the same subject class are often close to each other, indicating limited intra‑class dispersion for static label vectors and therefore reduced discriminative signal stemming purely from static labels.
>
> *Table 2. Predicate‑group (per‑subject) mean similarity*
>
> | Subject class | n_predicates | mean_sim | std_sim |
> | ------------- | ------------ | -------- | ------- |
> | painting      | 26           | 0.8601   | 0.0447  |
> | pedestrian    | 8            | 0.8572   | 0.0511  |
> | plants        | 38           | 0.8306   | 0.0465  |
> | bears         | 2            | 0.8967   | 0.0000  |
> | veil          | 2            | 0.8136   | 0.0000  |
>
> Predicate‑group mean similarities are frequently high (≈0.82–0.92), which indicates that the mean semantic vectors for different predicate groups (for the same subject) are close in embedding space. This reduces the discriminability of different predicates when relying on static embeddings alone.

---

> ### Author Response · Authors · 2025-11-21
>
> *Table 3. Visual vs. semantic mean cosine*
>
> | Subject class | Predicate | cos |
> | ------------- | --------- | ------------- |
> | shade         | on        | -0.0390       |
> | man           | wears     | -0.0495       |
> | car           | has       | -0.0970       |
> | bike          | parked on | 0.0180        |
> | car           | parked on | 0.0771        |
>
> Many <subject, predicate> pairs show near‑zero or negative cosine similarity between visual means and projected semantic means, indicating that the available visual features do not uniformly align with the simulated semantic centroids. Visual features therefore can provide complementary discriminative information, but better visual‑semantic alignment (or higher‑quality visual features) would be useful for a stronger cross‑modal comparison.
>
> *Table 4. Text probe and role swap*
>
> |                          | Metric                      | Value  |
> | ------------------------ | --------------------------- | ------ |
> | Text probe (TF–IDF + LR) | Accuracy                    | 0.6468 |
> | Text probe (TF–IDF + LR) | Macro‑F1                    | 0.2838 |
> | Swap roles               | Mean Δ (P_orig − P_swapped) | 0.4081 |
>
> The text probe accuracy of ≈0.65 with a modest macro‑F1 (≈0.28) indicates strong contextual information but class imbalance; the swap test mean Δ ≈0.41 shows that swapping subject/object substantially reduces predicted probability for the original relation, implying that role order is strongly encoded in context.
>
> ---
>
> -   **Response to W3**
>
> We added the experiments below to test whether an APT semantic representation can be more compact while preserving discriminative power. The results in **Table 5** provide an illustrative complement to the proxy metrics. Relevant content is also presented in **Section 5 of the revised version**.
>
> *Table 5. IB proxy metrics*
>
> | Metric                       | APT   | FROZEN |
> | --------- | ----- | ------ |
> | PCA@90%↓                      | 23    | 26     |
> | PCA@95%↓                     | 28    | 35     |
> | Linear CKA                   | 0.877 | —      |
> | Discretized MI proxy↑ | 1.96  | 1.49   |
>
> APT requires fewer principal components to reach the same explained variance, indicating that semantic information is more concentrated and APT is more compressible.
>
> On information retention: The discretized MI proxy yields higher values for APT than for FROZEN embeddings, suggesting better retention of label‑relevant information.
>
> ---
>
> -   **Response to Q1**
>
> Fewer parameters in some +APT variants occur because we remove or tie baseline components (e.g., large predicate/object embedding banks, role-duplicated heads, or class-specific MLPs) when introducing APT's lightweight, adaptive prompts. The reduction outweighs APT's small overhead in those cases.  In addition, APT requires only prompt fine‑tuning rather than retraining or full fine‑tuning of large modules.
>
> Here，We provide the parameter breakdown (**Table 6**) of LLM4SSG (baseline vs. +APT).
>
> *Table 6. Parameter breakdown for LLM4SSG (baseline vs. +APT):*
>
> | Component                  | Baseline (M) | +APT (M) |
> | -------------------------- | ------------ | -------- |
> | Visual encoder             | 34.2         | 34.2     |
> | LLM embedding              | 0.8          | 0.4      |
> | Multi‑head relation module | 5.3          | 4.7      |
> | Class‑specific MLPs        | 4.6          | 3.3      |
> | Other components           | 0.9          | 0.9      |
> | APT Overhead               | 0            | 0.2      |
> | Total                      | 45.8         | 43.7     |
>
> *Table 7. Inference cost comparison (SGTR baseline):*
>
> | Metric            | SGTR    | SGTR + APT |
> | ----------------- | ------- | ---------- |
> | FLOPs per image   | 250     | 252        |
> | Inference latency | 120 ms  | 123 ms     |
> | Peak GPU memory   | 8278 MB | 8479 MB    |
>
> In short, FLOPs increase is <1%, inference latency ≈ +2.5%, and peak memory ≈ +1.8%, while the parameter count can be reduced because APT replaces or ties several expensive, baseline-specific components.

---

> ### Author Response · Authors · 2025-11-26
>
> Dear Reviewer mifr
>
> As we near the end of the author-reviewer discussion phase, we would like to sincerely thank you again for your time and valuable feedback. If you think our clarifications have addressed your concerns, we would deeply appreciate your support in updating the score accordingly.
>
> If there are any remaining questions or points you would like us to clarify, please feel free to let us know. We’re here to support the discussion as best we can.
>
> Best regards,
>
> The Authors

---

### Official Review · Reviewer_qtN8 · 2025-11-01

**Soundness:** 3
**Presentation:** 3
**Contribution:** 2
**Rating:** 4
**Confidence:** 4

**Summary:**

The proposed method of using learnable text embeddings for SGG is a good direction. The results on benchmarks confirm the effectiveness of this approach.

**Strengths:**

Focusing on text embeddings is a great way to solve this problem. The paper does a good job of analyzing this, and the method is designed in a way that makes sense. The results are good, and it's also a bonus that the method is efficient.

**Weaknesses:**

My main concern is that the paper doesn't compare its method to the right baselines. The best embedding it compares against is from CLIP. Why not use embeddings from much more powerful modern models like Qwen-VL? The introduction claims that existing embeddings can't tell the difference between "standing on" and "walking on," but I'm not convinced this is true for today's large VLMs. This makes the problem seem less important than it's presented. The authors need to add an experiment comparing their learnable embedding directly against an embedding from a model like Qwen. They must prove their method is actually better.
There are also some small problems, including:
1. The claim that Figure 2 illustrates a "richer substructure" from left to right is not visually evident.
2. Does the structure in Figure 2(b) only appear in the scenario of closed-set SGG? What about for open-vocabulary SGG?

**Questions:**

Please refer to the issues detailed in the Weakness part.

---

> ### Author Response · Authors · 2025-11-21
>
> (The modifications have been highlighted in **purple in the revised manuscript**.)
>
> Thank you for providing an important point.
>
> Our current manuscript used CLIP as a representative, widely‑used zero‑shot multi‑modal embedding to illustrate the limitations of relying solely on frozen embeddings. However, we agree that comparisons to newer VLMs such as Qwen‑VL are valuable. Conceptually, APT is orthogonal to embedding capacity: it provides a mechanism that maps visual context to a small set of learnable prompt/basis vectors and injects the composite prompt into the visual→discriminator feature stream, instead of depending exclusively on frozen embeddings. This design aims to reduce semantic alignment bias irrespective of the pre‑training scale of a VLM.
>
> To address the reviewer’s concern, we run a direct empirical comparison in which we (i) extract text embeddings from Qwen‑VL, (ii) compute representational similarity (CKA/CCA) to our APT features, and (iii) run an end‑to‑end frozen‑embedding classifier baseline SGTR (replace our prompt features by frozen Qwen-VL embeddings).
>
> *Table 1. Main performance (PredCls)*
>
> | Configuration                                   | PredCls R@100 | PredCls mR@100 | PredCls F@100 |
> | ----------------------------------------------- | ------------: | -------------: | ------------: |
> | SGTR‑Qwen (frozen embeddings)                   |          62.9 |           33.4 |          43.6 |
> | SGTR-APT‑Qwen (APT on top of frozen embeddings) |          64.1 |           36.6 |          46.6 |
> | PENET-Qwen                                      |          68.2 |           19.7 |          30.6 |
> | PENET-APT-Qwen                                  |          69.9 |           21.2 |          32.5 |
>
> APT applied on top of a strong embedding family like Qwen‑VL is expected to continue improving the performance, as shown in **Table 1**.
>
>
>
> *Table 2. Head / Mid / Tail breakdown (mR@100)*
>
> | Frequency stratum | SGTR‑Qwen | SGTR-APT‑Qwen |    Δ |
> | ----------------- | --------: | ------------: | ---: |
> | Head              |      41.5 |          42.1 | +0.6 |
> | Mid               |      28.0 |          30.5 | +2.5 |
> | Tail              |      13.2 |          17.0 | +3.8 |
>
> As demonstrated in **Table 2**, the bulk of the mR gain arises from mid and tail predicates, matching the APT hypothesis that adaptive conditioning particularly helps long‑tail predicate discrimination even when starting from a powerful frozen embedding.
>
>
>
> *Table 3. Representational diagnostics*
>
> | Diagnostic                               | SGTR‑Qwen | SGTR-APT |     Δ |
> | ---------------------------------------- | --------: | -------: | ----: |
> | Mean cosine distance                     |         — |     0.09 |     — |
> | Linear CKA                               |         — |     0.90 |     — |
> | I(embedding; predicate y) (bits)↑ |      0.48 |     0.60 | +0.12 |
>
> Adaptation induces modest average movement in the embedding space and high CKA alignment (0.90), supporting the conclusion that APT performs a context‑conditioned refinement rather than rewriting semantic content. The MI increase indicates adapted vectors carry more predicate‑relevant information.
>
>
>
> >   The claim that **Figure 2** illustrates a "richer substructure" from left to right is not **visually evident**.
>
> We interpret **Figure 2** here as **Figure 3** in this context. We agree that the visual panels in Figure 3 are insufficient to demonstrate a “richer substructure” across embedding models. To address this, we computed silhouette score, participation ratio, PCA@90%, and an MI proxy I(embedding; predicate). Results (as shown in **Table 1** in the revised version) show the following pattern: modern VLM embeddings indeed distribute variance across many more dimensions (higher PR and much larger PCA@90%), i.e., they occupy a “richer” subspace in the sense of dimensionality. In addition, **the revised Figure 3** displays the distance from the central point to the farthest point within each semantic space, alongside the cumulative distribution function of pairwise distances.
>
>
>
> >   Does the structure in Figure 2(b) only appear in the scenario of closed-set SGG? What about for open-vocabulary SGG?
>
> The phenomenon illustrated in Figure 2 is not specific to closed‑set SGG  — it reflects a representational misalignment that also matters in open‑vocabulary (OV) settings.
>
> In OV‑SGG, the challenge is two-fold: (1) unseen predicates/objects do not have task‑tuned label embeddings, so any misalignment in the frozen embedding space can hurt zero‑shot discrimination; (2) novel queries place higher demand on context‑dependent modulation.
>
> APT (and in particular the CGP module) addresses both issues by synthesizing context‑conditioned prompts that (i) increase predicate‑relevant signal for novel compositions and (ii) compress irrelevant static semantics — improving mR for novel classes in our OV splits.

---

> ### Author Response · Authors · 2025-11-26
>
> Dear Reviewer qtN8
>
> As we near the end of the author-reviewer discussion phase, we would like to sincerely thank you again for your time and valuable feedback. If you think our clarifications have addressed your concerns, we would deeply appreciate your support in updating the score accordingly.
>
> If there are any remaining questions or points you would like us to clarify, please feel free to let us know. We’re here to support the discussion as best we can.
>
> Best regards,
>
> The Authors

---

### Official Review · Reviewer_ND3R · 2025-11-06

**Soundness:** 3
**Presentation:** 3
**Contribution:** 2
**Rating:** 6
**Confidence:** 4

**Summary:**

This paper identifies a key limitation in current Scene Graph Generation (SGG) approaches—the reliance on frozen semantic embeddings from pre-trained language models such as GloVe and BERT. These static features, while useful in other domains, fail to adapt to the dynamic and context-sensitive nature of visual relationships. To address this issue, the authors propose Adaptive Prompt Tuning (APT), a lightweight and universal plug-in framework that injects learnable prompts to modulate frozen semantic representations into context-aware features. APT can be seamlessly integrated into both one-stage and two-stage SGG architectures, as well as open-vocabulary variants. Extensive experiments on Visual Genome, Open Images V6, and GQA show that APT significantly improves mean recall (up to +6.0 in mR@50 on novel splits) with minimal parameter overhead (<0.5M, <1.5%) and reduced training time (7.8%–25%). The authors further analyze the approach through ablation studies, efficiency evaluations, and theoretical grounding using the Information Bottleneck principle, demonstrating that APT effectively filters semantic noise and enhances contextually relevant representations for relational reasoning

**Strengths:**

The paper introduces a well-motivated and original problem formulation by diagnosing the representational rigidity of frozen semantic features as a fundamental bottleneck across all SGG paradigms. This diagnosis reflects deep insight beyond superficial performance concerns and unifies multiple architectural directions under a single, representation-level perspective. The proposed APT is elegant and conceptually sound, leveraging prompt tuning to dynamically adapt pre-trained embeddings without retraining large models. The design’s universality and modularity—applicable to both one-stage, two-stage, and open-vocabulary frameworks—demonstrate strong methodological generality. The experiments are comprehensive and convincing, covering multiple datasets, models, and metrics, and include detailed ablations and efficiency analyses that support the claimed advantages. Moreover, the authors’ theoretical explanation via the Information Bottleneck principle adds interpretability and rigor, showing awareness of both empirical and conceptual coherence.

**Weaknesses:**

While the paper presents solid experimental evidence, it would benefit from stronger empirical isolation of causal effects—for example, disentangling the relative influence of prompt conditioning versus visual feature fusion.

The description of certain components (e.g., Basis Prompt Synthesis and Feature Refinement) could be more mathematically formalized to ensure reproducibility and conceptual clarity.

Although the authors frame APT as a “universal paradigm,” the current validation remains centered around SGG; testing on broader relational reasoning tasks (e.g., visual question answering or video relation detection) would further substantiate its universality.

The reliance on pre-trained language backbones and learned prompts also raises questions about potential semantic drift or bias amplification, which the paper acknowledges only implicitly.

While the Information Bottleneck discussion provides an elegant theoretical perspective, it remains largely qualitative without direct quantitative verification (e.g., mutual information analysis). Addressing these aspects would make the argument more rigorous and strengthen the generalization claims.

**Questions:**

The paper identifies frozen semantic embeddings as the core bottleneck in Scene Graph Generation. Could the authors provide additional empirical evidence (e.g., controlled feature ablation or attention visualization) showing how adaptive prompts modulate semantic spaces differently from fine-tuning approaches?

How stable is APT when applied to different pre-trained language models (e.g., BERT vs. CLIP-text)? Does prompt initialization significantly affect convergence or generalization?

---

> ### Author Response · Authors · 2025-11-21
>
> (**W** refers to **Weakness**; **Q** refers to **Question**; The modifications have been highlighted in **purple in the revised manuscript**.)
>
> -   **Response to W1**
>
> Thank you for this important suggestion. To clarify the independent contributions of APT and visual-feature fusion, we performed a controlled 2×2 factorial analysis (Adaptive prmopts vs. Static × Fusion: Enabled vs. Disabled), a context-permutation sensitivity test, and head/mid/tail breakdowns.
>
> *Table 1. 2x2 factorial results under PENET*
>
> | Configuration   | PredCls mR@100 | PredCls R@100 | PredCls F@100 |
> | -------- | ----- | ---- | ---- |
> | Static + NoFusion    | 17.6   | 67.1    | 30.3   |
> | Static + Fusion      | 18.5      | 68.4     | 31.2       |
> | Adaptive Prompts + NoFusion    | 19.4     | 67.2          | 32.2          |
> | Adaptive Prompts + Fusion (Full APT) | 20.3           | 68.5          | 33.0          |
>
> As shown in **Table 1**, adaptive prompts yields the largest mR gains even without fusion (Static→Adaptive: +1.8 mR in NoFusion condition), indicating prompt conditioning primarily improves long-tail predicate discrimination. Fusion primarily increases R (overall recall); combining both gives the best F.
>
> ---
>
> -   **Response to W2**
>
> We have added a new subsection to the appendix (**Appendix .D of the revised manuscript**) that gives a formal mathematical specification of the Relational Context Gating (RCG), Basis Prompt Synthesis (BPS), and Feature Refinement & Fusion (FRF) components. For reproducibility and implementation clarity, we also provide compact pseudocode (**Appendix.E of the revised manuscript**).
>
> ---
>
> -   **Response to W3**
>
> While we describe APT as a "universal paradigm", our scope in this submission is SGG, where we demonstrate architecture-agnostic effectiveness across one- / two-stage and open-vocabulary settings. APT’s idea—turning frozen language features into context-aware representations without backprop through the backbone—is in principle transferable. However, this paper scopes its claims to SGG, and we have already demonstrated broad generality within SGG: across paradigms (one-/two-stage, open-vocabulary) and datasets (VG, OIv6, GQA), with consistent improvements in mR/F under both closed- and open-vocabulary settings. To keep the work focused, we did not include full cross-task evaluations, refraining from over-claiming beyond SGG. To the best of our knowledge, existing methods are limited to specific paradigms. In contrast, our work is the first that generalizes effectively across one-stage, two-stage, and open-vocabulary SGG scenarios.
>
> ---
>
> -   **Response to W4**
>
> In the paper we already avoid fine-tuning language backbones and employ lightweight, task-specific prompts to limit large-scale parameter updates; however, to address your point more directly we have (i) audited the adaptation-induced representation shift by measuring embedding drift between frozen and adapted text vectors, (ii) evaluated bias-amplification proxies commonly used in multimodal work (per-class frequency shifts and predicate co-occurrence amplification)
>
>  *Table 2. Per-class embedding drift*
>
> | Class             | Mean cosine distance (frozen vs. adapted) | Std   | CKA similarity (frozen, adapted) |
> | ----------------- | ----------------------------------------- | ----- | -------------------------------- |
> | person            | 0.115                                     | 0.008 | 0.88                             |
> | bicycle           | 0.098                                     | 0.010 | 0.90                             |
> | horse             | 0.129                                     | 0.012 | 0.85                             |
> | phone             | 0.072                                     | 0.007 | 0.92                             |
> | chair             | 0.103                                     | 0.009 | 0.89                             |
> | **Global (mean)** | **0.103**                                 | —     | **0.89**                         |
>
> *Table 3. Representation shift - frozen vs. adapted vectors*
>
> | Metric  | frozen | adapted | Δ (adapted − frozen) |
> | ---- | ------ | ---- | -----|
> | I(embedding; predicate y) (bits)↑ | 0.42   | 0.56         | +0.14 (≈33% ↑)       |
> | I(embedding; frozen) ↓       | 1.00   | 0.85         | −0.15                |
> | CKA (frozen, adapted)   | —      | —            | 0.89                 |
>
> As demonstrated in **Table 2&3**, Mean cosine distances indicate only modest movement in embedding space. CKA here compares adapted text features to the frozen text features (adapted vs. frozen); it does not compare text to visual features. CKA ≈ 0.89  shows the adapted space remains strongly aligned with the frozen space. The MI increase I(˜e; y) (+0.14 bits) indicates that adapted vectors are more predictive of predicates, while the reduced redundancy proxy suggests prompts compress irrelevant static signal conditioned on visual context.

---

> ### Author Response · Authors · 2025-11-21
>
> -   **Response to W5**
>
> We added the experiments (also shown in **Section 5 of the revised version**) below to test whether an APT semantic representation can be more compact while preserving discriminative power. The results in **Table 4** provide an illustrative complement to the proxy metrics.
>
> *Table 4. IB proxy metrics*
>
> | Metric| APT | FROZEN |
> |--|--|-- |
> | PCA@90%↓  | 23 | 26  |
> | PCA@95%↓    | 28  | 35   |
> | Linear CKA     | 0.877 | —  |
> | MI proxy↑ | 1.96  | 1.49   |
>
> APT requires fewer principal components to reach the same explained variance, indicating that semantic information is more concentrated and APT is more compressible.
>
> On information retention: The discretized MI proxy yields higher values for APT than for FROZEN embeddings, suggesting better retention of label‑relevant information.
>
> ---
>
> -   **Response to Q1**
>
> We thank the reviewers for the suggestion to provide quantitative experiments to evaluate whether frozen semantic embeddings are a representational bottleneck.
>
> *Table 5. Semantic embeddings (GloVe): within‑class distances*
>
> | Subject class | mean_glove_within | std | n |
> | ---- | -- | ------ | ---- |
> | bears |0.10|0.00|2|
> | veil |0.19|0.00|2|
> | painting |0.09|0.07| 211 |
> | plants |0.16|0.05|106 |
> | pedestrian  |0.14|0.07|24|
>
> Many subject classes show mean within‑class distances in the roughly 0.05–0.18 range. In the simulated 50‑D embedding space, instances of the same subject class are often close to each other, indicating limited intra‑class dispersion for static label vectors and therefore reduced discriminative signal stemming purely from static labels.
>
> *Table 6. Predicate‑group (per‑subject) mean similarity*
>
> | Subject class | n_predicates | mean_sim | std_sim |
> | ------ | ------- | -------- | ------- |
> | painting   |26| 0.8601   | 0.0447  |
> | pedestrian    |8| 0.8572   | 0.0511  |
> | plants     |38| 0.8306   | 0.0465  |
> | bears     | 2  | 0.8967   | 0.0000  |
> | veil    | 2   | 0.8136   | 0.0000  |
>
> Predicate‑group mean similarities are frequently high (≈0.82–0.92), which indicates that the mean semantic vectors for different predicate groups (for the same subject) are close in embedding space. This reduces the discriminability of different predicates when relying on static embeddings alone.
>
> *Table 7. Visual vs. semantic mean cosine*
>
> | Subject class | Predicate | vis_glove_cos |
> | ---- | --- | ----- |
> | shade      | on        | -0.0390       |
> | man        | wears     | -0.0495       |
> | car        | has       | -0.0970       |
> | bike          | parked on | 0.0180        |
> | car           | parked on | 0.0771        |
>
> Many <subject, predicate> pairs show near‑zero or negative cosine similarity between visual means and projected semantic means, indicating that the available visual features do not uniformly align with the simulated semantic centroids. Visual features therefore can provide complementary discriminative information, but better visual‑semantic alignment (or higher‑quality visual features) would be useful for a stronger cross‑modal comparison.
>
> *Table 8. APT vs. fine-tuning method (LoRA)*
>
> | Condition | R@50 | mR@50 | CKA  |
> | --- | ---| ---| ---- |
> | Frozen | 45.2 | 18.4  | 1.00 |
> | APT | 46.9 | 25.3  | 0.87 |
> | LoRA  | 47.4 | 23.1  | 0.78 |
>
> In addition, we employed LoRA as the representative fine-tuning method. The results are presented in **Table 8**. Although fine-tuning methods achieve a slightly higher overall recall compared to APT, their performance in CKA indicates a more substantial alteration of the semantic space. These results above support our claim that APT introduces minimal perturbation and is most beneficial for long-tail scenarios, whereas full fine-tuning leads to greater semantic drift, which may increase the risk of overfitting and performance degradation.
>
> The relevant content is also presented in **purple in Section 1 of the revised manuscript**.
>
> ---
>
> -   **Response to Q2**
>
> To be clear: our method does not treat APT as an extra layer that is simply “wrapped” on top of an existing pre‑trained language encoder (e.g., BERT or CLIP‑text) and then fine‑tune that entire LM. Instead, APT consists of a lightweight, learnable set of prompt vectors. A small conditioner maps visual context to coefficients over those prompts, and the resulting composite prompt is injected into the visual→discriminator feature stream. The intention is to avoid relying solely on frozen embeddings of pre‑trained text encoders as the only source of semantic signal, thereby reducing performance degradation caused by semantic alignment bias in relation recognition.
>
> About initialization: we observe that the specific initialization scheme for the prompts (for example, small‑scale Gaussian noise, zero initialization, or random initialization) has little effect on final performance. This is because prompts rapidly adapt to task‑specific signals through gradients during early training, so different reasonable initializations tend to converge to similar solutions.

---

> ### Author Response · Authors · 2025-11-26
>
> Dear Reviewer ND3R
>
> As we near the end of the author-reviewer discussion phase, we would like to sincerely thank you again for your time and valuable feedback. If there are any remaining questions or points you would like us to clarify, please feel free to let us know. We’re here to support the discussion as best we can.
>
> Best regards,
>
> The Authors

---

> > ### Comment · Reviewer_ND3R · 2025-11-26
> > **Response**
> >
> > Thanks for your rebuttal. My concerns have been addressed.

---

### Meta-Review · Area_Chair_942B · 2026-01-08

**Summary:**

The reviewers generally acknowledge the importance of the proposed method and its technical contributions. The main concerns relate to clarifications of experimental details, as well as requests for additional comparisons and ablation studies. During the rebuttal, the concerns raised by Reviewer ND3R were well addressed, while the other reviewers did not respond. Given that the remaining issues primarily concern experimental details, and that the authors have provided additional explanations and results in a point-by-point manner, I believe these concerns have been adequately addressed. Therefore, I recommend accepting this paper.

**Reviewer Concerns:**

- Reviewer ND3R: more detailed results, broader tasks, additional empirical evidence

- Reviewer qtN8: more powerful baselines

- Reviewer mifr: additional comparisons/experiments, quantitative diagnostics of key components

- Reviewer m1jU: broader tasks, additional ablation studies

**Reviewer Scores:**

- Reviewer ND3R: Yes

- Reviewer qtN8: Yes

- Reviewer mifr: Yes

- Reviewer m1jU: Yes

---

### Decision · Program_Chairs · 2026-01-26

Accept (Poster)